# Cryo-EM reveals the architecture of placental malaria VAR2CSA and provides molecular insight into chondroitin sulfate binding

Kaituo Wang [1,9], Robert Dagil [2,9], Thomas Lavstsen[2], Sandeep K. Misra [3], Charlotte B. Spliid[2,4], Yong Wang [5], Tobias Gustavsson[2,6], Daniel R. Sandoval[4], Elena Ethel Vidal-Calvo [2,6], Swati Choudhary[2], Mette Ø Agerbaek[2], Kresten Lindorff-Larsen [5], Morten A. Nielsen [2], Thor G. Theander[2], Joshua S. Sharp [3,7], Thomas Mandel Clausen[4], Pontus Gourdon [1,8✉] & Ali Salanti [2✉]

Placental malaria can have severe consequences for both mother and child and effective vaccines are lacking. Parasite-infected red blood cells sequester in the placenta through interaction between parasite-expressed protein VAR2CSA and the glycosaminoglycan chondroitin sulfate A (CS) abundantly present in the intervillous space. Here, we report cryo-EM structures of the VAR2CSA ectodomain at up to 3.1 Å resolution revealing an overall V-shaped architecture and a complex domain organization. Notably, the surface displays a single significantly electropositive patch, compatible with binding of negatively charged CS. Using molecular docking and molecular dynamics simulations as well as comparative hydroxyl radical protein foot-printing of VAR2CSA in complex with placental CS, we identify the CS-binding groove, intersecting with the positively charged patch of the central VAR2CSA structure. We identify distinctive conserved structural features upholding the macro-molecular domain complex and CS binding capacity of VAR2CSA as well as divergent elements possibly allowing immune escape at or near the CS binding site. These observations will support rational design of second-generation placental malaria vaccines.

[1] Department of Biomedical Sciences, University of Copenhagen, Copenhagen, Denmark. [2] Centre for Medical Parasitology at Department for Immunology and Microbiology, Faculty of Health and Medical Sciences, University of Copenhagen and Department of Infectious Disease, Copenhagen University Hospital, Copenhagen, Denmark. [3] Department of BioMolecular Sciences, University of Mississippi, Oxford, MS, USA. [4] Department of Cellular and Molecular Medicine, University of California, San Diego, La Jolla, CA, USA. [5] Structural Biology and NMR Laboratory & Linderstrøm-Lang Centre for Protein Science, Department of Biology, University of Copenhagen, Copenhagen, Denmark. [6] VAR2Pharmaceuticals, Ole Maaloesvej 3, Copenhagen, Denmark. [7] Department of Chemistry and Biochemistry, University of Mississippi, Oxford, MS, USA. [8] Department of Experimental Medical Science, Lund University, Lund, Sweden. [9] These authors contributed equally: Kaituo Wang, Robert Dagil. ✉email: pontus@sund.ku.dk; Salanti@sund.ku.dk

**P**lasmodium falciparum is the most lethal human malaria parasite. During the erythrocytic stage, the parasites infect, multiply, rupture and re-infect red blood cells. Infected erythrocytes are effectively removed from the blood by splenic filtering. However, to avoid splenic clearance the parasites express proteins on the surface of the infected erythrocytes that anchors these cells to the host vasculature. The binding is mediated by members of a family of proteins called *Plasmodium falciparum* Erythrocyte Membrane Protein 1 (PfEMP1)[1]. In each parasite genome, PfEMP1s are encoded by ~60 different *var* genes, including *var2csa*[2,3]. The transcription of *var* genes is regulated in a mutually exclusive manner, ensuring that only one PfEMP1 variant is expressed on the surface of an infected erythrocyte at any given time[4]. VAR2CSA allows infected erythrocytes to sequester in the placental vasculature, causing placental malaria[5]. While infections in pregnant women are often clinically silent, they do cause maternal anemia and significantly impair fetal growth. Thus, placental malaria is estimated to result in 900,000 low birth weight deliveries each year in Africa. It has long been known that in malaria-endemic areas pregnant women have a higher risk of being infected with malaria than non-pregnant women[6]. This is due to the establishment of the placenta, which creates a new niche for binding of infected erythrocytes during pregnancy. Early work from Fried and Duffy showed that the malaria parasites accumulate in the placenta by binding to a glycosaminoglycan of chondroitin sulfate type A (CS) abundantly present in the placenta[7]. Later, it was identified that infected erythrocytes binding to placental chondroitin sulfate A (plCS) expressed a PfEMP1 gene named *var2csa*[5]. After exposure to malaria during pregnancy, women develop antibodies against VAR2CSA, which inhibit parasites from binding to placental tissue, and protect against placental malaria during subsequent pregnancies. Therefore, VAR2CSA is utilized for the development of vaccines protecting women against placental malaria, and two vaccines are currently in clinical development[8,9].

VAR2CSA is embedded in the erythrocyte membrane through a single C-terminal transmembrane spanning segment. Compared to other PfEMP1s, the ~310 kDa VAR2CSA ectodomain is relatively conserved among parasites[10]. It is composed of a short N-terminal segment and six Duffy Binding-Like (DBL) domains (DBL1-DBL6) unique to VAR2CSA. Several of the DBL domains are linked by complex inter-domain (ID1-3) regions with limited or no homology among PfEMP1s yielding a DBL1-ID1-DBL2-ID2-DBL3-DBL4-ID3-DBL5-ID4-DBL6 structure[10].

The ligand for VAR2CSA is Chondroitin Sulfate A, a glycosaminoglycan (GAG), which is composed of repeating disaccharide units of N-acetyl-D-galactosamine (GalNAc) and glucuronic acid (GlcA). CS can be attached to proteins as non-branched linear GAG sidechains, and thereby form proteoglycans. In the placenta the CS modified major receptor is syndecan-1[11,12]. The CS disaccharide units can be modified by the addition of sulfate groups, such as sulfation of the hydroxyl groups at C2 of GlcA, and/or C4 (4-*O*-sulfation) and C6 (6-*O*-sulfation) of GalNAc. Several studies suggest that VAR2CSA interacts with a CS saccharide of a length between 12-16 saccharides consisting primarily, but not exclusively of 4-*O*-sulfated units[13,14]. Interestingly, the specific CS signature recognized by VAR2CSA is not only found in placenta, but also on almost all malignant tumors[13,15]. This has enabled specific targeting of solid tumors as well as capture of circulating tumor cells using recombinant VAR2CSA[16].

For many years structural information was scarce and relied on single VAR2CSA DBL domains structures[17–19], then followed low-resolution envelope structures of full length proteins determined by small-angle X-ray scattering providing insight into domain architecture though shed limited light into ligand binding[20]. The first insight into the ligand binding of VAR2CSA was provided by Bewly et al. in 2020[21] suggesting a model with two distinct CS binding pores that could accommodate a CS chain. Recently Ma et al.,[22] published the cryo-EM structures of VAR2CSA FCR3 in its apo-form and VAR2CSA NF54 in complex with CS in agreement with the presence of a CS binding groove in ID1-ID2a. However, until now, vaccine design has relied on empirical screening of large panels of N- and C-terminally truncated recombinant proteins. The approach defined DBL2 and flanking regions as the minimal CS binding region[23,24]. Thus, the recombinant VAR2CSA proteins in clinical development are the PrimVac construct comprising the ~100 kDa DBL1-DBL2, and the PamVac construct comprising the ~70 kDa ID1-ID2a region[8,9]. These VAR2CSA forms maintain high affinity binding to CS and induce high levels of inhibiting antibodies toward the homolog parasite variants but they are less effective in inhibiting the binding of heterologous variants. This probably reflects that both vaccine proteins include regions that are diverse in sequence among VAR2CSA variants, and therefore likely to confer antigenic diversity. High-resolution structural insight and identification of the residues involved in the CS interaction is therefore needed for the development of more effective vaccines to protect against placental malaria.

Here, we describe the structure of the VAR2CSA ectodomain, displaying an intricate core assembly of N-terminal domains, followed by more loosely arranged C-terminal domains. The structure shows that DBL2 is central for charge-complementation for CS binding, exposing a series of positively charged residues on the surface. Using molecular docking of CS oligosaccharides, we show that atypical DBL features of VAR2CSA DBL2 domains form the structural basis for CS binding. By fast photochemical oxidation of proteins (FPOP) analyses of VAR2CSA in complex with placental CS and mutational analyses, we validate the structural and surface exposed regions involved in CS binding and demonstrate that the groove can encompass a CS oligo of around 13 saccharides. Structural loops, polymorphic among VAR2CSA variants, found near the CS binding site may have evolved to escape antibody recognition. These observations enable design of new VAR2CSA-based placental malaria vaccines, providing hope for a vaccine eliciting a broadly reactive, parasite binding inhibiting antibodies.

## Results and discussion

**Expression and buffer selection of full length VAR2CSA.** The extracellular region from the N-terminal methionine (M1) to amino acid F2649 representing the entire ectodomain of the FCR3 VAR2CSA variant was expressed in baculovirus transfected insect cells as a secreted monomeric protein and purified by immobilized nickel affinity chromatography (Fig. 1a). A selection of different post purification formulation buffers and salts were screened and analyzed using size-exclusion chromatography. By comparing neutral pH buffers with either NaCl or KCl as electrolyte, we observed that KCl containing buffers yielded a more compact VAR2CSA configuration (elution at 9.9 ml in NaCl and 12.1 ml in KCl) (Fig. 1b). To further test the different buffers, we measured the binding kinetics between the CS proteoglycan CSPG and VAR2CSA using a Quartz-Crystal Microscale biosensor. From the fitted association and dissociation rates, we obtained VAR2CSA binding affinities, which were 10-fold stronger in KCl ($k_D = 0.1$ nM) than in NaCl buffer ($k_D = 1.4$ nM) (Fig. 1c). This indicates that the VAR2CSA conformation and associated CS-binding are highly dependent on the electrolyte present, which may be important during cellular trafficking to the erythrocyte membrane and for ligand binding in the placenta milieu. As the binding to CSPG in NaCl buffer showed a 1:2 stoichiometry

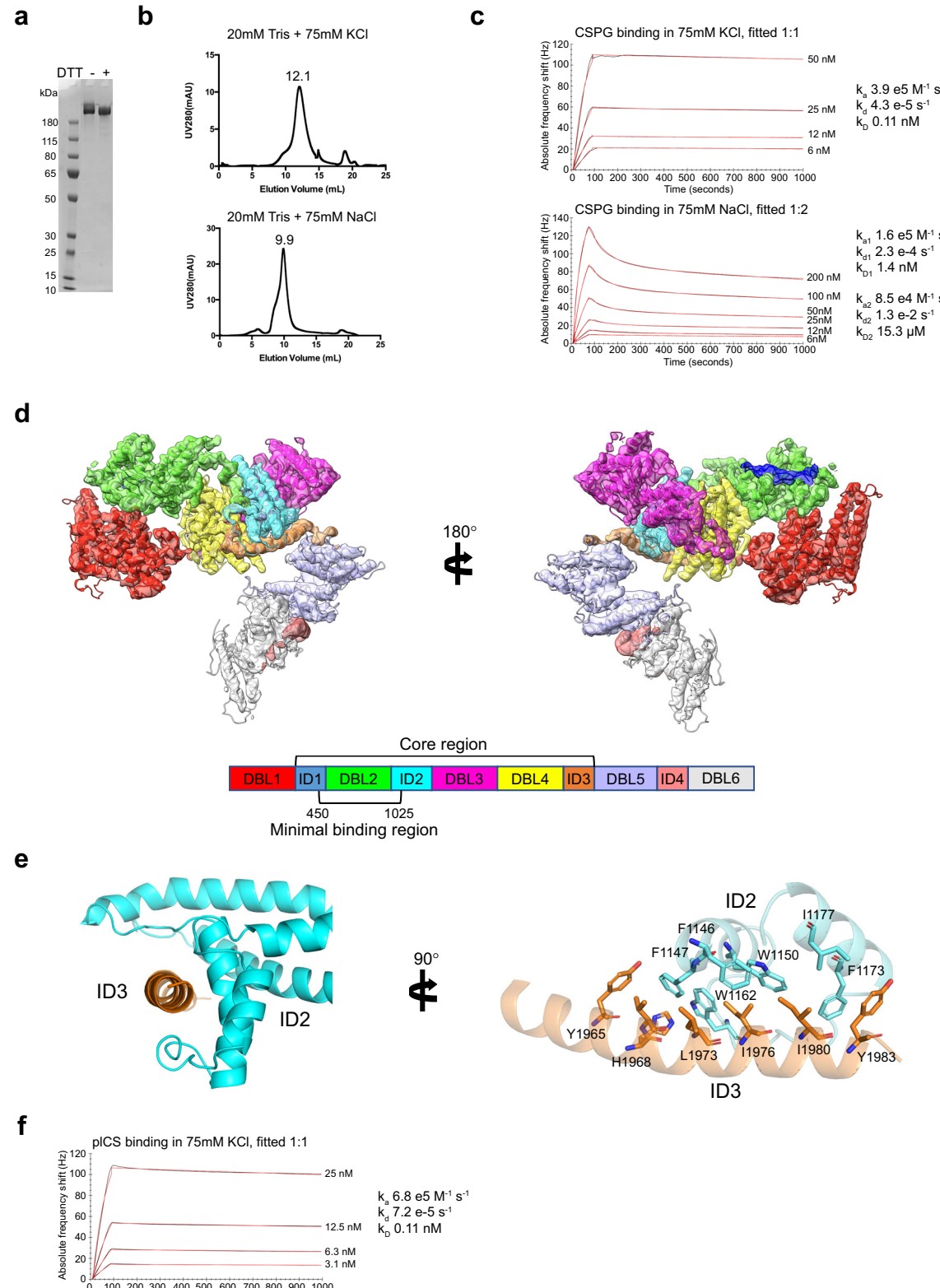

**Fig. 1 Apo-structure of the ectodomain of VAR2CSA. a** SDS-PAGE non-reduced/reduced of full-length VAR2CSA (307 kDa). A gel was run after purification of VAR2CSA before the protein was aliquoted and stored at −80 °C until used. **b** Gel filtration profile of VAR2CSA in different buffers, top panel KCl, bottom panel NaCl. **c** QCM biosensor interaction between CSPG decorin and VAR2CSA, top panel KCl ($k_D$ 0.11 nM), bottom panel NaCl ($k_{D1}$ 1.4 nM). Black curves represent recorded data, red curves fitted data. **d** Overall structure of VAR2CSA, flipped 180° to the right. The separate domains are colored as shown in the overview below. The core region of the structure and minimal CS binding region (AA450-1025) are indicated in the legend. **e** α-helix from ID3 (orange) AA1955-1985 interaction with C-terminal part of ID2 (blue). To the right the structure is flipped 90° and side chains from interacting residues are shown as sticks and labeled. For clarity, helices are shown with 50% transparency. **f** QCM biosensor interaction between pICS and VAR2CSA in buffer with KCl ($k_D$ 0.11 nM). Black curves represent recorded data, red curves fitted data. Source data are provided in source data file.

compared to a 1:1 in KCl buffer, the less compact VAR2CSA could introduce more unspecific binding or the proposed additional minor groove binding site is more exposed in NaCl buffer allowing CSPG binding. Structural analysis was performed in KCl buffer, as the highest affinity toward CSPG was observed.

**Cryo-EM structure of the VAR2CSA ectodomain.** Next, we determined the cryo-EM structure of the isolated VAR2CSA form generated from a map of overall 3.8 Å resolution. The final model represents the ectodomain of VAR2CSA, exhibiting a well-folded conformation with clearly defined DBL domains (Figs. 1d, S1a) and in agreement with previously published structures of DBL3, DBL4 and DBL6[17–19]. The region spanning ID1-DBL2-ID2-DBL3-DBL4-ID3 represents the core of the macromolecular structure, as highlighted by a compact structure with a high degree of inter-domain interactions (Fig. 1d). This observation aligns with the structural envelope proposed by SAXS analyses previously[24]. The structure discloses the fold of ID2, forming a separate domain consisting of a bundle of α-helices, a shape which together with ID3, a 30 residues long α-helix (residue 1955-1985), serves as a structural glue for the core. Notably, ID2 and ID3 interact, as the C-terminus of ID2 forms an open-mouth shape conformation, which interacts extensively with the ID3 α-helix (Figs. 1e, S2a, b). Underscoring the significance of the ID2-ID3 interface, VAR2CSA sequence conservation analysis indicates that the ID2 and ID3 interacting regions are highly conserved (Fig. S3a,b), suggesting a common fold for all VAR2CSA variants. The structure of the core is further maintained through interactions between several DBL domains and inter-domain-links. Specifically, dense contacts are detected between DBL2 and DBL4, ID2 and DBL4, ID2 and DBL3 as well as DBL3 and DBL4. The fold of the DBL3-DBL4 region is similar to the previously described fold (PDB-ID 4P1T)[17] (Fig. S4). DBL1 exhibits only a few inter-domain interaction sites with DBL2 and DBL4, leaving a cleft in-between the core and DBL1. Likewise, the C-terminal DBL5 and DBL6 are more separated from the core, linking the protein to the membrane in a physiological setting. The DBL5 and DBL6 form a rod-like structure with weak inter-domain contacts (Fig. 1d).

**VAR2CSA Cryo-EM structure in the presence of placental CS.** To shed further light on how VAR2CSA binds CS, we determined the cryo-EM structure of VAR2CSA supplemented with CS purified from placental tissue (plCS). The binding of VAR2CSA to plCS in KCl buffer was measured using QCM biosensor, confirming interaction in the low nanomolar range (Fig. 1f). Comparing the complex structure to the structure solved in the absence of CS, the new data yielded a 3.1 Å resolution map overall, extending to 2.8 Å in the core region (Fig. S1b,c,d). By contrast, DBL1 and DBL5/6 were less well resolved compared to the apo structure (Figs. S1, S5). Flexibility of these domains is further supported by two separately generated maps based on different subsets of particles, showing the same core structure, but different conformations of DBL5/DBL6 (Figs. S1e, S6). Thus, it is possible that the peripheral regions of the structure become flexible and potentially displaced upon plCS binding. Conversely, these data indicate that plCS does not induce any conformational changes in the core, as structures of the DBL2-ID2-DBL3-DBL4-ID3 domains were highly similar in the presence or absence of plCS. Although plCS binding affinity to VAR2CSA is in the low nanomolar range, we did not observe any additional density that could represent ligand binding in the presence of plCS. This may be due to the intrinsic heterogeneity of plCS prepared from its natural placental source. Interestingly, analysis of the surface electrostatics instead revealed a single strongly positively patch,

situated in a groove that spans through the VAR2CSA core which could encompass a negatively charged CS oligosaccharide. The positive patch is located in the cleft between DBL1 and the core, in-between DBL1, DBL2 and DBL4 and includes the experimentally defined minimal CS binding region, consisting of DBL2 and flanking inter-domain-stretches[24]. Thus, our cryo-EM data were in agreement with a key role of the positively charged surface region around the DBL2 domain for binding of CS. We also noted that the DBL1-core cleft, which extends a possible binding region beyond the electropositive area, to a large extent is formed by DBL4. This is of interest as DBL4 was previously a lead candidate for vaccine development, because recombinant DBL4 protein can elicit adhesion-blocking antibodies[25]. Interestingly, we previously demonstrated that to induce functional inhibitory antibodies the DBL4 domain boundaries required the inclusion of the ID3 region, which now appears close to the proposed CS binding groove[25].

**VAR2CSA-specific DBL elements confer both CS binding and immune escape.** We noted that key features of the VAR2CSA structure are established through distinct adaptations of the DBL domains. DBL domains of both PfEMP1 and non-PfEMP1 proteins from different species of *plasmodia* are all built on same basic fold[26] with core helices comprising three subdomains 1-3 (S1-3). Subdomains 1 and 2 are folded together and typically, cysteine bridges span within but not between S1-2 and S3. S1 has no conserved secondary structure, but S2 and S3 are formed by 4 and 3 helical bundles, respectively. The key residues maintaining this fold are relatively conserved and for most part found in short helix-forming sequence stretches defined as homology blocks 1-5 (HB1-5)[10] (Fig. S7). Previously characterized PfEMP1-protein receptor interactions are mediated by residues in, or adjacent to, HB1[27–30]. Atypical for DBL domains in general, the HB1 helix in VAR2CSA DBL2 is broken up mid-helix by an insertion forming a flexible loop (Figs. 2a, S7). This loop is glycine-rich, but polymorphic between VAR2CSA variants varying from 4-11 amino acid in length. Compared to DBL3 and DBL4, the C-terminal part of HB1 in DBL2 forms part of the highly positively charged surface in the core of the structure (Fig. 2b). Thus, while irregular in its makeup, the DBL2 is similar to other PfEMP1 domains, in that the proposed key ligand-binding site is located in the C-terminal part of HB1. The insertion of a structural and sequence variable region near this site may reflect evolutionary selection for antigenic variable elements providing protection and escape from antibody recognition.

At the other end of the CS binding groove, there is a Lysine-rich loop flanked by the WIW-motif that is conserved in all VAR2CSA DBL2s (Fig. 2c, d). This region contains a tryptophan conserved in all DBL domains (W558), which serves the function of stabilizing the overall DBL structure through interactions with another tryptophan in HB2 (W766) of the DBL2 (Fig. 2d). The region is structurally and sequence polymorphic among different DBL domains, but particularly enriched in positively charged amino acids in VAR2CSA DBL2 (Fig. 2b, d). Interestingly, the same region together with positively charged residues near the HB1 C-terminal mediate the binding to glycophorin A in the glycan-binding channels formed by dimerization of the *P. falciparum* EBA-175 protein, when merozoites invade erythrocytes[31]. Altogether, these observations support the hypotheses that CS binds a positively charged groove sloping through the VAR2CSA core structure, and that the lysine-rich region in the DBL2 N-terminus loops provides an ideal albeit antigenically diverse interaction point to the groove.

In DBL4, the HB5 is also unusually elongated and broken up mid-helix. Here, an 8-15 amino acid long loop is inserted, presenting a conserved Arginine-Lysine pair possibly binding the

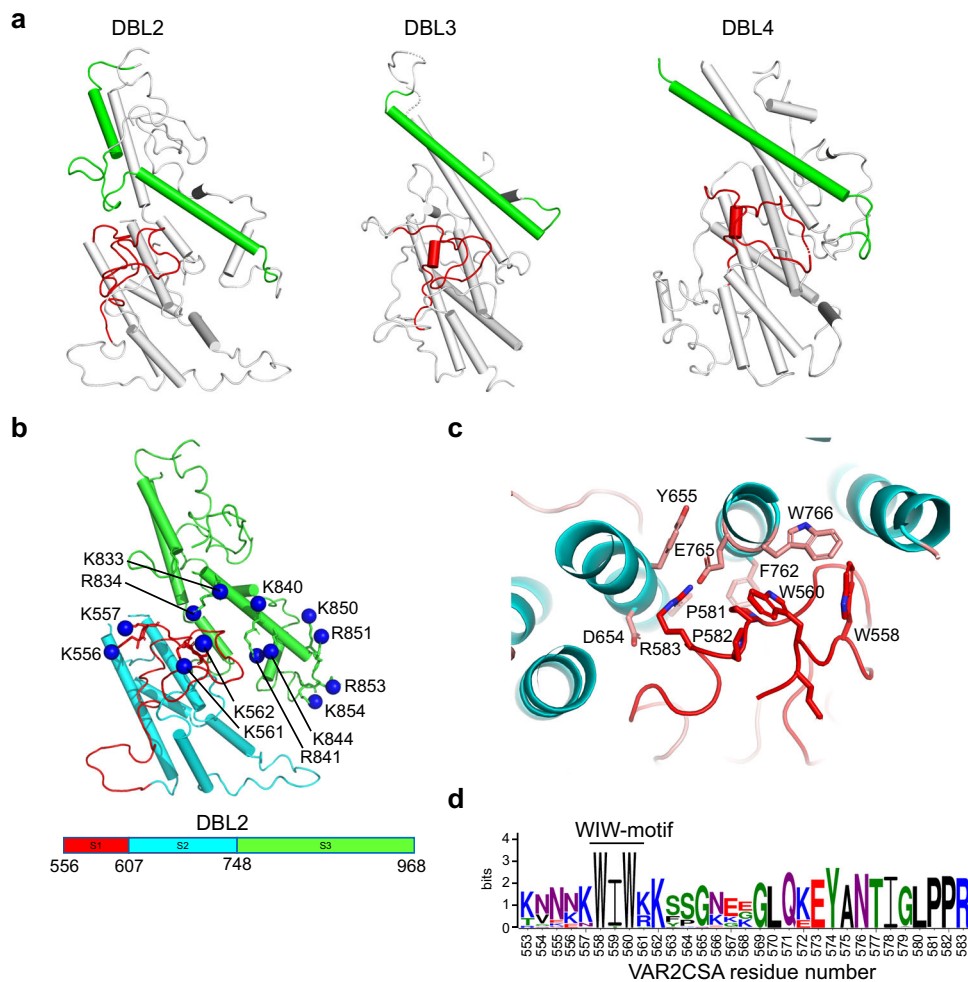

**Fig. 2 Structural analysis of DBL2 and CS binding groove. a** Comparison of domains DBL2, DBL3 and DBL4. The red loop shows the "WIW-motif"-loop from each domain, the green α-helix shows HB1. In DBL2, an additional loop in HB1 introduces a kink in the α-helix. **b** DBL2 domain shown as cartoon, the red loop shows S1 (HB4 and "WIW-motif"-loop), blue α-helices show S2 (HB3-HB5) and green α-helices show S3 (HB2-HB1). S1-S3 residue boundaries and colors are shown in legend below. The distal nitrogen atom of Lys/Arg residues side chains in binding groove are shown as blue spheres and labeled according to residue number. **c** Structural details of the interaction between "WIW-motif" and HB2/HB3 in DBL2. Interacting side chains are shown as sticks and are labeled with residue numbers. **d** LOGO conservation analyses of "WIW-motif"-loop AA553-583 in *var2csa* variants. Residue number from VAR2CSA sequence shown below.

protruding CS chain, and a sequence-variable flexible stretch of 2-7 amino acids, possibly shielding the one end of the CS-binding groove from antibody recognition.

**Mapping CS-protected sites by FPOP-MS.** To gain additional support for the proposed CS binding groove, we expressed the previously defined CS binding region as a recombinant DBL1-ID2 protein. Using this DBL1-ID2, we applied Fast Photochemical Oxidation of Proteins (FPOP) with and without plCS complexed to the protein. Following treatment of the VAR2CSA or plCS-VAR2CSA complex with chymotrypsin, oxidation differences in the respective peptides were compared. The sequence coverage and oxidation coverage achieved is shown in Figs. S7 and S8, with extracted ion chromatograms for each oxidized peptide shown in Fig. S9. We identified various regions primarily in ID1, DBL2 and ID2 that exhibited protection from solvent upon binding to plCS (Figs. 3a, S8). The protection of the ID1, DBL2 and ID2 domains support the notions of either a flexible binding site with partial or dynamic engagement of multiple residues of the protein, and/or a binding-induced conformational change resulting in widespread shielding of multiple parts of

these domains from solvent. The peptide that showed the largest absolute reduction in oxidation after plCS binding was peptide 543-558, with oxidation of this peptide occurring on W558 and the segment 547-CK-548 (Fig. S10). This peptide and the C-terminal adjacent peptide, also protected by plCS, spans the above-mentioned lysine-rich region N-terminal to DBL2 HB4 (Fig. S7). In addition, the FPOP analyses showed protection of the HB1 DBL2 region. Together, these observations support the role of the positively charged DBL2 surface and HB1 groove (Supplementary Movie 1) as the CS binding region.

**Mutagenesis of DBL1-ID2 for CS binding assay.** Unusual for DBL domains, the DBL2 N-terminal region contains not one but two conserved tryptophan residues, creating a conserved "WIW-motif" (Fig. 2d). In the FCR3 VAR2CSA variant studied here, this WIW-motif is flanked by several lysine residues 555KKK**W**I**W**KK562. To assess the importance of the fold of this region for CS binding, we produced a recombinant DBL1-ID2 protein with a 555KKK**A**I**A**KK562 substitution to alter the configuration of the loop. The integrity and purity of the wild-type and mutant protein was analyzed by SDS-PAGE (Fig. 3b) and

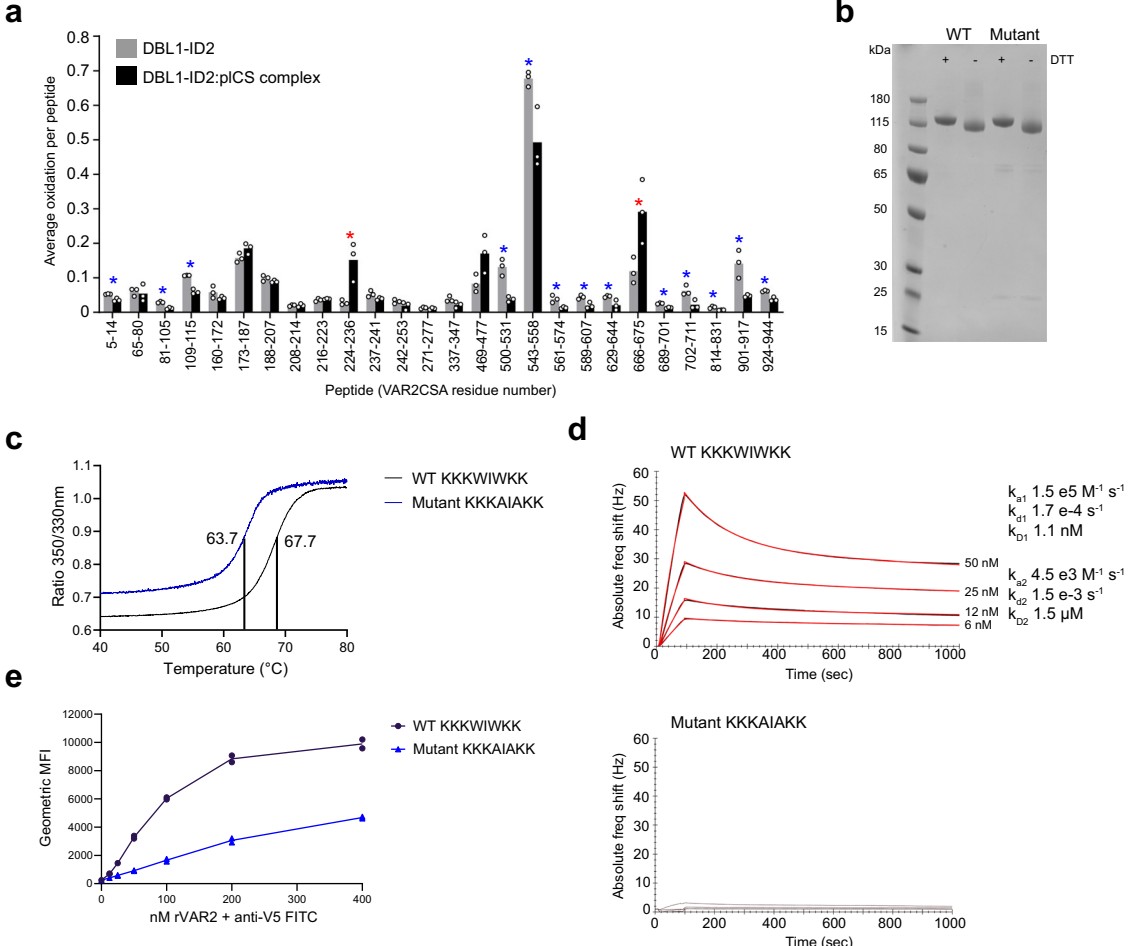

**Fig. 3 Hydroxyl radical protein foot printing and CS binding analysis of DBL1-ID2 mutant. a** FPOP analysis of placental CS binding to wild-type DBL1-ID2. 13 chymotryptic peptides in VAR2CSA were found to exhibit significant protection from oxidation upon binding to placental CS (blue asterisks, $p < 0.05$, two-tailed Student's $t$ test). Two peptides (red asterisks, $p < 0.05$, two-tailed Student's $t$ test) exhibit significant exposure. Peptides not detected as oxidized in either condition are not shown. **b** SDS-PAGE non-reduced**/**reduced of DBL1-ID2 wild-type (KKKWIWKK) and mutant (KKKAIAKK). The gel shown is from one purification, the mutant DBL1-ID2 was purified three times and the wild-type protein more than three times. **c** $T_m$ determination by nanoDSF melting curves of wild-type ($T_m$ 67.7 °C) and mutant ($T_m$ 63.7 °C) DBL1-ID2. Region between 40 °C and 80 °C is shown. **d** QCM biosensor kinetic measurements of wild-type (top, $k_{D1}$ 1.1 nM) and mutant DBL1-ID2 (bottom, no binding) to decorin CSPG. **e** Flow cytometry binding analyses of lung cancer cell line A549 to wild-type and mutant DBL1-ID2. Source data are provided in source data file.

measurement of melting temperature. The melting temperature of the mutant showed a decrease of ~4 °C compared to the wild-type protein, indicating local structural changes in line with W558 being involved in local structural integrity. However, a transition from folded to unfolded was still observed for the mutant (Fig. 3c). When assessing the CSPG interaction using a biosensor, binding of the mutated protein was almost abolished compared to wild type (Fig. 3d). Similar results were obtained when binding was evaluated through binding to CS expressed a lung carcinoma cell line evaluated by flow cytometry (Fig. 3e). These results underscore the importance of the WIW-motif in positioning the positively charged residues for CS binding.

**Modeling the VAR2CSA-CS complex.** Guided by the positively charged surface region, we generated a docking model of VAR2CSA in complex with a CS 20-mer oligosaccharide, for which the stability was validated by unbiased atomistic molecular dynamics simulations. Strikingly, the docking model showed a near perfect fitting of the CS oligo chain along the positively charged groove (Fig. 4a, b, c). The MD simulations

revealed a stable core region with about 7 disaccharide units (Fig. 4d), in good agreement with previous experimental estimation[14]. The simulations also suggested significant flexibility of DBL1 and DBL5/6, consistent with the relatively poor EM density in these regions (Fig. S11). Combined with the fact that the DBL1 domain appeared more flexible in the CS-bound cryo-EM structure, it may indicate that DBL1 acts as a cover, which protects the CS binding region from antibody recognition prior to ligand binding, and is displaced once CS is bound. A recent publication by Ma et al.,[22] shows the cryo-EM structures of VAR2CSA FCR3 in its apo-form and VAR2CSA NF54 in complex with CS. The similarity of the VAR2CSA FCR3 apo-structures by Ma et al., and the one presented here is high, with RMSD in the core region (AA residue 450–2022) of 1.38 Å. The complex structure from Ma et al., confirms the involvement of both the WIW-motif and the C-terminal part of HB1 in DBL2 in CS binding. In summary, the cryo-EM analysis of VAR2CSA demonstrates the overall architecture and the domain arrangement achieved through intricate interactions between DBL domains. This assembly facilitates the binding of long chain CS molecules through a positively charged electrostatic groove sloping through the protein core (Fig. 4a). The center of this groove can

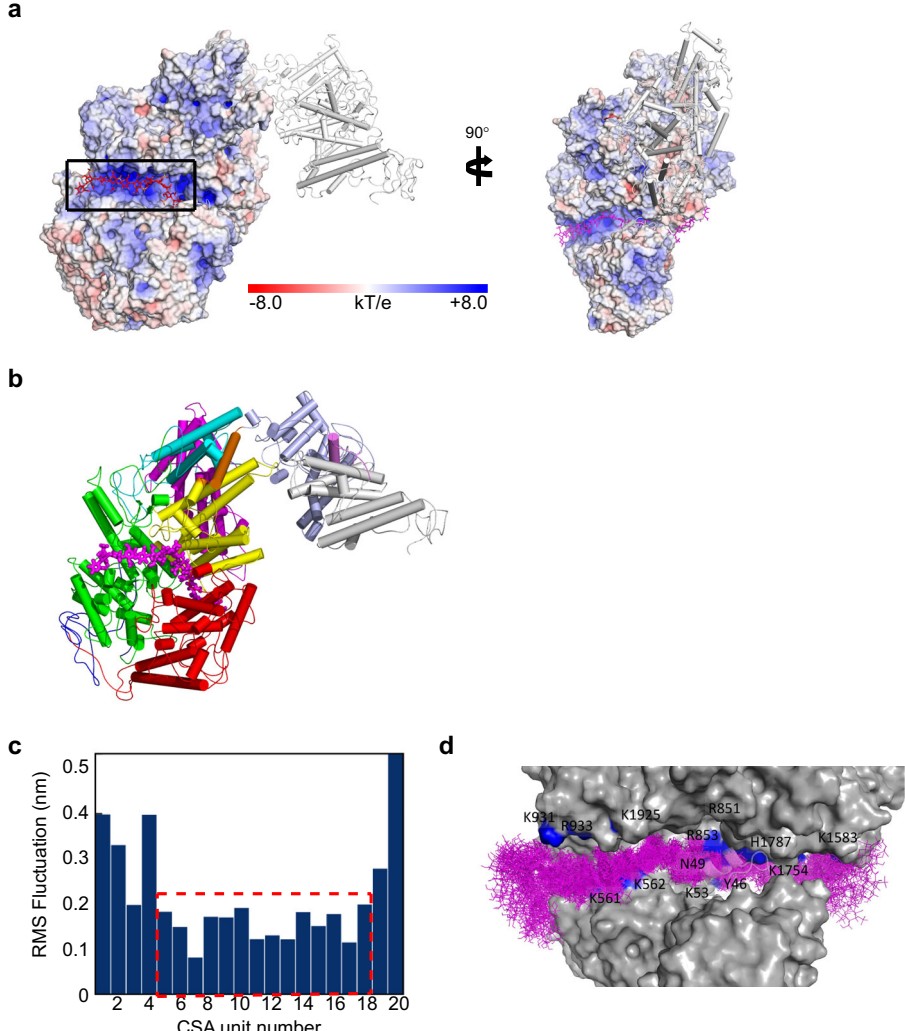

**Fig. 4 Docking model of VAR2CSA in complex with a CS 20-mer. a** The docking structure of VAR2CSA in complex with CS 20-mer. The electrostatic potential surfaces of VAR2CSA with the minimal binding region and binding groove, which is highlighted by a black box. A surface potential (kT/e) color bar is shown. The CS molecule is represented by sticks in red. The DBL5/6 domains are shown in white cartoon. The right panel shows the structure turned 90°. **b** Docking structure of VAR2CSA shown as cartoon with domains colored as in Fig. 1d. The CS molecule is shown as sticks in magenta. **c** Conformational flexibility of CS ligand in the binding groove of VAR2CSA. Boxed area shows the core part of the CS chain with relatively low RMSF values, corresponding to roughly seven disaccharide units. **d** Binding groove of VAR2CSA with an ensemble of CS 20-mer with 70 conformers sampled in a 70 ns MD simulation with an interval of 1 ns. CS interacting residues from VAR2CSA are labeled with residue numbers.

accommodate a CS oligosaccharide of around 7 disaccharide units (Fig. 4c). Our data suggest that the peripheral DBL1, 5 and 6 domains are displaced upon ligand interaction, whereas the core region maintains its structure. The binding site is surrounded by flexible loops containing polymorphic VAR2CSA sequences possibly evolved to provide escape from immune recognition. Malaria vaccines based on the CS binding region of VAR2CSA have failed to induce anti-adhesion responses that cross-react with heterologous variants of VAR2CSA. The high-resolution structure and insight into the binding region provide hope for future development of an intelligently designed cross reactive placental malaria vaccine, targeting conserved features of the CS binding groove.

## Methods

**Protein cloning, expression and purification**. The VAR2CSA extracellular fragment starting from the N-terminal methionine M1 to amino acid residue F2649, located prior the putative transmembrane region, was amplified from codon optimized FCR3 (GenBank™ accession no. GU249598) and cloned into baculovirus vector pAcGP67-A (BD Biosciences) including a V5 and 6xHis-tag at the C-terminal. The construct was co-transfected into Sf9 cells to generate virus particles used to infect high-five insect cells (as previously described[24,32]). Expression was induced for 2 days and supernatant cleared by centrifugation at $10,000 \times g$ for 15 min at 4 °C. The supernatant was concentrated and buffer-exchanged into PBS pH 7.4 buffer using a 50,000 MWCO hollow-fiber filter. Imidazole to a final concentration of 60 mM was added and then loaded onto a 5 ml HisTrap HP (Cytiva) column. The protein was eluted using a linear gradient toward PBS with 300 mM imidazole pH 7.4. Purity and homogeneity were verified by SDS-PAGE and protein was aliquoted and stored at −80 °C. VAR2CSA wild-type construct DBL1-ID2 was expressed in SHuffle T7 express competent *E. coli* cells (NEB) and purified from the cytoplasm using HisTrap HP (Cytiva), followed by HiTrap SP HP (Cytiva) cation exchange chromatography. DBL1-ID2 $^{555}$KKK**A**I**A**KK$^{562}$ mutant was generated using Thermo Fisher site directed mutagenesis kit and verified by sanger sequencing. Protein expression and purification was performed similar to wild-type DBL1-ID2.

**Cryo-EM sample preparation**. For apo sample preparation, a frozen aliquot of VAR2CSA was subjected to size-exclusion chromatography in 20 mM Tris pH 7.5, 125 mM KCl. The generated protein peak was concentrated to 8.6 mg/ml and fluorinated Fos-Cholin8 (FC8) was added to the protein sample to a final concentration of 1 mM immediately before freezing. Quantifoil 1.2/1.3 holy carbon grids were glow-discharged with a Leica Coater ACE 200 for 30 s using 5 mA current. Cryo-EM grids were prepared with a Vitrobot Mark IV operated at 100%

humidity and 4 °C. A 3.5 μl aliquot of purified protein was applied to the grids, incubated for 5 s, blotted 3.5 s and plunge frozen into liquid ethane. The sample with plCS was prepared by incubating VAR2CSA with plCS (purified as previously described[33,34]) in a 1:7 mass ratio for 30 min at 18 °C. The sample was chondroitinase ABC treated with 300mU enzyme for 1.5 h and further purified using a Superdex200 increase 10/300GL column equilibrated in 20 mM Tris pH 7.5 and 75 mM KCl. Peak fractions were concentrated to 0.5 mg/ml. The cryo-freezing conditions were identical except no FC8 was added.

**Cryo-EM data collection and processing**. The apo sample cryo-EM datasets were collected on a Titan Krios electron microscope (FEI) operated at 300 kV with a Falcon3 direct detector camera. A total of 8081 movies were recorded under linear mode at pixel size of 0.832 Å and a total dose of 60 e/Å2 spread over 19 frames. Cryo-EM data were processed using cryosparc[35]. The initial processing steps were full-frame-motion corrected and the CTF estimated with ctffind4[36] (wrapped in cryosparc). Blob particle picking was done in a subset (2166) of the micrographs and particles were extracted and 2D classified to prepare a template-based particle selection. All template selected particles were re-extracted using local-motion correction with dose-weighting and using a box size of 440 pixels. A total of 940,807 auto-picked particles were subjected to several rounds of reference-free two-dimensional class averaging to clean-up clearly defective particles. The cleaned-up particle set was processed with the standard cryosparc workflow, including ab-initio model reconstitution, multiple rounds of heterogeneous refinement and non-uniform refinement iterations. The final map was calculated using a subset of 102,676 particles to an overall resolution of 3.82 Å, with the best parts stretching to about 3.6 Å (Fig S5).

The CS complex sample dataset (in total 5234 movies) was collected at pixel size of 0.832 Å in Falcon3 counting mode. The total dose was 40e/Å2 over 40 frames. Using a similar data processing strategy as for the apo structure, a total of 1,022,972 particles were extracted and a final subset of 266,774 particles were used to produce a map of overall 3.1 Å resolution, with the more well-resolved portion reaching about 2.8 Å (Fig. S6). The final dataset could be further classified into three different subsets, with almost identical core-structure but displaying variance in the DBL5/6 domains. Further local refinement within a mask that covers the DBL5/6 region only improved the resolution of DBL5/6 to about 4 Å using one of the three subsets.

**"Imbalanced" hetero refinement**. During data processing, the hetero-refinement step in Cryosparc was inspired by the "random-phase 3D Classification" method described by Gong et al.[37]. As example, following the first round the non-uniform (NU) refinement, the final map was low-pass-filtered to resolution of 40 Å using Volume tools in Cryosparc. In the hetero-refinement step, the NU-refinement map and the 40Å-low-pass-filtered map was used as two initial models and the "Initial resolution" parameter was set to 20 Å. This generated two "imbalanced" sub-set of particles with different size, where the low-pass-filtered map served as a "Trash can" to filter out lower quality particles that have poorer agreement with the high-resolution map. By adjusting the low-pass-filter resolution, initial resolution, box-size, and number of refinement iterations, usually 5–25% of low-quality particles were removed in each round, and the resolution/map quality increased until reaching consensus (Supplementary Figs S5 and S6, Table S2).

**Model building and refinement**. Published crystal structures of DBL3/4 (PDB-ID 4P1T)[17] and DBL6 (PDB-ID 2Y8D)[38] were directly docked into the corresponding region with good fit. For DBL domains lacking published structures (DBL1/2/4/5), the initial models were generated using the SWISS-MODEL online server with the corresponding sequences and PDB-ID 2WAU as a template. In the 3.1 Å map, the core structure (residue AA556-1985) was built de novo with high confidence except for some flexible loop regions. The model building was done iteratively using COOT[39] and phenix_real_space_refine of the Phenix software package[40]. Secondary structure restraints and Ramachandran restraints were also imposed during refinement. The resolution of DBL1 and DBL5 was insufficient for de novo model building, and these domains were refined using homology model and later MD-assisted model refinement. The quality of the model was assessed using Molprobity[41] (see Table S1 for statistics). We have submitted three structures: the apo structure from DBL1 to ID3 (EMD-12017, PDB-ID 7B52), CS structure from DBL1 to ID3 (EMD-12018, PDB-ID 7B54) and a CS structure local refined of DBL5/6 (EMD-12477, PDB-ID 7NNH). The overall structure (Fig. 1d) was generated using the 3.8 Å apo map. All other structure figures were generated using the CS complex structure.

**Molecular docking and MD simulations of the VAR2CSA in complex with CS**. The refinement models of core domains together with the homology model of low resolution domains were subsequently threaded by adding missing residues using Modeller9.18[42] to construct a full-length model of VAR2CSA. Missing residues within and between domains were modeled as unstructured loops. The full-length model was then used as a template to fit into the Cryo-EM density map of the apo and CS VAR2CSA structures, respectively, using molecular dynamics flexible fitting (MDFF) method with secondary structure, cis-peptide and chirality restraints to prevent overfitting[43]. MDFF was performed using an implicit solvent with a

scaling factor of the map potential, $g = 0.3$. The models were refined by multiple rounds of manual adjustment in PyMol and optimization in MDFF, as well as energy minimization in Gromacs (version 2019)[44] using CHARMM36m force field[45]. CS was modeled as a 20mer which was subsequently used to dock with the highly positively charged regions of VAR2CSA using HADDOCK 2.4[46]. The docked structures were further refined by energy minimization using Gromacs. Electrostatic surfaces were analyzed using APBS plugin in PyMol.

The docking model of VAR2CSA in complex with CS was subsequently placed into a periodic cubic box with sides of 18.7 nm solvated with TIP3P water molecules containing $K^+$ and $Cl^-$ ions at 0.1 M, resulting in 193,271 molecules (620,728 atoms) in total. The CHARMM36m force field was used for the protein. Force field parameters for CS were generated using the Glycan Modeler module in the CHARMM-GUI web interface[47]. Neighbor searching was performed every 20 steps. The PME algorithm was used for electrostatic interactions with a cut-off of 1.2 nm. A reciprocal grid of 160 ×160 x 160 cells was used with 4th order B-spline interpolation. A single cut-off of 1.2 nm was used for Van der Waals interactions. Temperature coupling was done with the Nose-Hoover algorithm. Pressure coupling was done with the Parrinello-Rahman algorithm. The hydrogen mass repartitioning technique[48] was employed with a single LINCS iteration (expansion order 4), allowing simulations to be performed with an integration time step of 4 fs. MD simulations of VAR2CSA complex were performed using Gromacs 2019.5. The interactions between VAR2CSA and the bound CS were analyzed by GetContacts scripts (https://getcontacts.github.io/). The flexibility of VAR2CSA domains and CS was analyzed using Gromacs rmsd and rmsf tools.

**Attana kinetic measurements**. Kinetic analysis of VAR2CSA binding to CSPG decorin and plCS was performed on a quartz crystal microbalance biosensor (Attana A200, Attana AB). Decorin CSPG was immobilized on a LNB carboxyl gold-coated sensor chip by amine coupling using S-NHS and EDC. For plCS immobilization, a LNB sensor chip was first coated with streptavidin (50 μg/ml) followed by an injection of 50 μg/ml biotinylated plCS. A shift in baseline verified plCS immobilization. The sensor chips were stabilized at 25 μl/min and 22 °C in 20 mM tris pH 7.5 with either 75 mM KCl or 75 mM NaCl and VAR2CSA was used as analyte in a two-fold dilution series from either 200 nM or 50 nM to 6.25 nM or 3.1 nM, respectively. For DBL1-ID2 wild-type and the KKKAIAKK mutant, the binding kinetics were measured in PBS pH 7.4. After each protein injection, the sensor surface was regenerated by 0.1 M NaOH. Buffer injections were subtracted from each binding curve and fitted to a 1:1 or 1:2 binding model using TraceDrawer (Ridgeview Instruments AB).

**NanoDSF**. Protein stability was assessed using NanoDSF instrument (Nanotemper Prometheus NT.48). DBL1-ID2 wild-type and the KKKAIAKK mutant were loaded into capillaries and heated by 1 °C/min from 20 °C to 90 °C. The fluorescence of 330 nm and 350 nm was monitored during heating and the ratio used to calculate $T_m$ when 50% of the protein was unfolded.

**Fast photochemical oxidation of proteins (FPOP)**. Glutamine, sodium phosphate, and catalase were purchased from Sigma-Aldrich (St. Louis, MO). LCMS-grade formic acid, water, acetonitrile, hydrogen peroxide and adenine were obtained from Fisher Scientific (Fair Lawn, NJ). Fused silica capillary was purchased from Molex, LLC (Lisle, IL). Sequencing grade modified trypsin was purchased from Promega (Madison, WI).

A final concentration of 2 μM DBL1-ID2 protein was incubated in 10 mM sodium phosphate buffer in the presence or absence of 2 μM placental chondroitin sulfate at pH 7.8 for 1 h. FPOP was performed as described previously[49]. Briefly, 20 μl of protein sample mixture containing 1 mM adenine, 17 mM glutamine and 100 mM hydrogen peroxide was flowed through a fused silica capillary with I.D. 100 μM. The sample was irradiated by a pulsed laser beam from a Compex Pro 102 KrF excimer laser at 248 nm wavelength (Coherent, Germany). The laser fluence at the intersection with the fused silica capillary flow cell was calculated to be ~13.1mJ/mm2/pulse. Laser repetition rate was set at 20 Hz and the sample flow rate was set at 16 μl/min to ensure a 15% exclusion volume between irradiated volumes. The samples were immediately quenched in a microcentrifuge tube containing 25 μl of quench mixture (0.5 μg/μl H-Met-NH2 and 0.5 μg/μl catalase). The adenine hydroxyl radical dosimetry readings were measured at 265 nm in a Nanodrop (Thermo Scientific) to ensure all the samples were exposed to equivalent amounts of hydroxyl radical[50]. The adenine reading difference of VAR2CSA FPOP in the absence of CS was $0.167 \pm 0.006$ and in the presence of CS was $0.177 \pm 0.006$ indicating that all the samples have comparable hydroxyl radicals. Each experimental condition, DBL1-ID2 with or without plCS, was performed in triplicate for statistical analysis.

A final concentration of 100 mM tris, pH 8.0 containing 1 mM CaCl2 and 5 mM DTT was added to the FPOP samples and incubated at 95 °C for 15 min to reduce and denature the protein. The sample was immediately cooled on ice for 2 min. Chymotrypsin with 1:20 ratio of chymotrypsin:protein was added and incubated at 25 °C for 16 h with rotation. The digestion reaction was stopped by the addition of 0.5% trifluoroacetic acid. The samples were analyzed on a Dionex Ultimate 3000 nano-LC system coupled to an Orbitrap Fusion Thermo Scientific (San Jose, CA). Samples were first injected via autosampler onto a 300 μM id X5 mm PepMap 100,

5 μm (Thermo Scientific) C18 trapping cartridge, then back-eluted onto an Acclaim PepMap 100 C18 nanocolumn (0.75 mm × 150 mm, 2 μm, Thermo Scientific). Peptides were separated using gradient of solvent A (0.1% formic acid in water) and solvent B (0.1% formic acid in acetonitrile) at a flow rate of 0.30 μl/min. The gradient consisted of 2–10% solvent B over 3 min, increasing to 35% B over 25 min, ramped to 95 % B over 3 min, held for 3 min, and then returned to 2% B over 2 min and held for 7 min. The spray voltage was set to 2,400 volts, and the temperature of the heated capillary was set to 300 °C. The data were collected in positive ion mode and full scans were collected in Orbitrap with resolution of 60,000 and MS2 scans were detected in ion trap. Full MS scans were acquired from m/z 250 to 2000 followed by eight subsequent MS2 CID scans on the top eight most abundant peptide ions. The peptides were fragmented by collision-induced dissociation CID with an isolation width of 3 m/z units.

Peptides were identified using Byonic version v2.10.5 (Protein Metrics). The Byonic search parameters included all possible major oxidation modifications as variable modifications and the enzyme specificity was set to cleave the protein after tyrosine, phenylalanine, tryptophan, and leucine. The peak intensities of the unoxidized peptides were used to search manually for oxidized peptides in the LC-MS data using accurate mass measurement and relative retention time shift compared to the unmodified peptide. The intensities of each unoxidized peptide and their corresponding oxidation products observed in LC-MS were used to calculate the average oxidation events per peptide in the sample as previously reported[51]. Briefly, peptide level oxidation was calculated by adding the ion intensities of all the oxidized peptides multiplied by the number of oxidation events required for the mass shift (e.g., one event for +16, two events for +32) and then divided by the sum of the ion intensities of all unoxidized and oxidized peptide masses as represented by Eq. (1).

$$P = [I(+16)oxidized\ X1 + I(+32)oxidized\ X2 + I(+48)oxidized\ X3 + \dots /[I unoxidized + I(+16)oxidized + I(+32)oxidized + I(+48)oxidized\dots] \quad (1)$$

where $P$ denotes the oxidation events at the peptide level and $I$ values are the peak intensities of oxidized and unoxidized peptides.

Where possible, CID MS/MS fragmentation was used to identify the major site (s) of oxidation within chromatographically resolved peptide oxidation products for peptides that show significant changes in oxidation upon binding plCS. Changes in the abundance of the identified chromatographically resolved peptide oxidation product relative to all versions of that peptide were measured to determine if that particular oxidation product changed upon plCS binding.

**Cell cultures.** A549 cells were cultured in DMEM medium. The media were acquired from Sigma Aldrich and supplemented with GlutaMaxTM, 10% fetal bovine serum and 1% penicillin-streptomycin. A549 cells were regularly detached using Trypsin-EDTA Solution (59417, Sigma-Aldrich) for passaging and all cells were sustained at 5% $CO_2$ and 37 °C.

**Flow cytometry.** Protein binding to A549 was tested by flow cytometry. A549 cells were detached with StemPro AccutaseTM Cell Dissociation Reagent (A1110501, Gibco). Cells were incubated with a twofold dilution of V5-tagged WT or mutant rVAR2 (400–12.5 nM) for 30 min at 4 °C, followed by an incubation with 500x diluted Anti-V5 FITC (Invitrogen 46-0308) for detection. Gating strategy showing forward and side-scatter is shown in Fig S12. Samples were processed on an LSR II flow cytometer (BD) and results were analysed using FlowJoTM software (BD Life Sciences).

**Sequence analysis.** A total of 3737 sequences annotated with VAR2CSA-specific DBL domains were extracted from the var database, varDB PF3K[52]. Of these, 1388 spanned across the NTS-DBL6 domain region. To assess sequence diversity, amino acid alignments were constructed for each domain separately using all sequences spanning a VAR2CSA domain using Muscle and subsequently edited by hand[53]. From these alignments, an alignment spanning NTS-DBL6 of the VAR2CSA ectodomain was constructed from those sequences spanning NTS-DBL6. Sequence LOGOs were made using WebLogo 3[54].

**Reporting summary.** Further information on research design is available in the Nature Research Reporting Summary linked to this article.

## Data availability

All data supporting the findings in the manuscript are available from the corresponding author upon reasonable request. The atomic coordinates and electron microscopy data have been deposited in the RCSB Protein Data Bank and in the Electron Microscopy Data Bank under the following entries: VAR2CSA FCR3 apo-structure DBL1-ID3 (PDB-ID 7B52, EMD-12017), VAR2CSA FCR3 plCS complex DBL1-ID3 (PDB-ID 7B54, EMD-12018) and VAR2CSA FCR3 plCS complex DBL5-DBL6 (PDB-ID 7NNH, EMD-12477). The raw LC/MS FPOP data have been made available on the ProteoeXchange server under accession number PXD024154 (extracted ion chromatograms are shown in Fig S9 and S10). The var gene sequences used in the work can be found at ftp://ftp.sanger.ac.uk/pub/project/pathogens/Plasmodium/falciparum/PF3K/varDB/. Source data are provided with this paper.

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

## Acknowledgements

We would like to express our deep gratitude to Benjamin Jacobsen, Andreas Frederiksen, Elham Alijazaeri and Susanne Lücking Nielsen, for their excellent technical assistance. CPR for use of nanoDSF instrument. S.K.M. and J.S.S. acknowledge support for this work from the National Institute of General Medical Sciences, National Institutes of Health (P41GM103390 and R01GM127267). K.L-L. and Y.W. acknowledge access to computational resources from the Danish National Supercomputer for Life Sciences (Computerome) and the ROBUST Resource for Biomolecular Simulations (supported by the Novo Nordisk Foundation), and financial support from the Lundbeck Foundation BRAINSTRUC initiative. KTW acknowledge access to computational resources from the Danish National Supercomputer for Life Sciences (Computerome). KTW has received support from Danish Research Councils and Lundbeck foundation. AS, TL have received support from the Danish Research Councils, ERC, Lundbeck and Carlsberg Foundations. MØA from Innovation Foundation Denmark. PG was financed by the Lundbeck and the Knut and Alice Wallenberg Foundations as well as by The Independent Research Fund Denmark and the Swedish Research Council.

## Author contributions

Protein expression and purification, flow cytometry, biosensor analyses were designed, performed and analyzed by R.D., T.G., E.E.V., S.C., M.Ø.A., M.A.N., A.S. FPOP analyses was performed and analyzed by S.K.M., J.S.S., T.M.C. and D.R.S. Cryo-EM and simulations were executed and analyzed by K.W., P.G., K.L.L., Y.W. Placental CS was purified by CS. Sequence analyses were achieved by TL and TGT. Study was designed by K.W., A.S., R.D.. Manuscript was written by A.S., T.G.T., P.G., K.W., T.L., R.D. All authors contributed to discussing the data and proof-reading the manuscript.

## Competing interests

A.S., T.G.T., M.A.N., T.M.C. are listed as co-inventors on a patent family covering the use of VAR2CSA to target and diagnose cancer. A.S. and T.G.T. are listed as co-inventor on a patent on using VAR2CSA as a prophylactic malaria vaccine during pregnancy. J.S.S. has financial interest in an early-stage company commercializing technologies for protein higher order structure analysis. The other authors have no conflicts of interest.
