## [Peer Review File · Nature Communications]

Reviewers' Comments:

Reviewer #1:

Remarks to the Author:

This manuscript describes the structures of VAR2CSA from FCR3 in the presence and absence of CS. It builds upon a lower resolution structure of the 3D7 variant published by others (BioRxiv April 2020; JBC Oct 2020) that identified the relative locations of DBL domains that created two pores involving DBL1-ID1-DBL2-ID2-DBL4. This earlier paper proposed a model for how VAR2CSA evaded immune detection and predicted the two CS binding sites, identified as high and low affinity in the current manuscript, and should be fully attributed for these contributions. For example, the structures described in this manuscript provide more details and support the previously published model that identified the relative location of domains and two pores/grooves as potential CS binding regions. This is contrary to your statement on lines 97-99.

A second more recent publication describing the structures of FCR and N strains by cryoEM has been published and all data is available (deposited in July 2020 and released Jan 13th 2021). Although now published, it occurred after this manuscript had been submitted and therefore a note to this effect needs to be added. However, the work described here is largely confirmatory and the two publications set the bar higher for the current manuscript. The data in this manuscript were deposited into PDB in Dec 2020 (7B52 and 7B54) and a summary table of data collection, refinement statistics included. Although this demonstrates that your results were finalized later, due to the closeness in time, I have not taken the results of this second paper into account when reviewing your manuscript. You may however, wish to incorporate the results more fully into your discussion as there are some interesting differences.

In general, the contributions of groups outside of your own have been, at best, overlooked in favor of self-citation. For example, the SAXS envelope described by Srivastava in a 2010 PNAS publication provided the first view of the ectodomain whereas the later work cited did not correctly assign relative positions ID and DBL domains described here. No one works in a vacuum and it is better to cite significant contributions appropriately. This needs to be fully addressed in the current manuscript. The data processing and model refinement statistics are poorer than I would expect. The percentage of residues in favored areas of the Ramachandran plot seems low especially considering Ramachandran restraints were imposed (line 370) and the rmsd in bond angles and lengths too broad suggesting geometric restraints were too loose. How do you account for this? Can this be improved? The protein produced by baculovirus is glycosylated. How do you think this impacts access to the CS binding surfaces? Could it have an effect?

It is surprising that CS-bound VAR2CSA increases the lability of DBL1. The description of interfacial residues involving DBL1 is vague and needs to be described in more detail. It would be helpful to color domains individually to better understand the relative contributions to the CS binding site (for example in figure 4b) and individual amino acids in the groove (figure 4d). This information is presented in linear form in Fig S7, so will be straightforward to apply to the structure.

In figure 1c, it is intriguing that binding measurements in the presence of 75 mM KCl results in 1:1 binding stoichiometry whereas in the presence of 75 mM NaCl, the stoichiometry is 1:2 since two pores were observed in the EM structures (neg stain and here) and earlier work by you and others show 1:1 stoichiometry in NaCl. Please account for this?

Crystal structures of DBL domains in complex with their cellular receptors identify similar regions of the DBL fold for receptor binding. Do the structures described in this manuscript explain why VAR2CSA does not bind these receptors?

Lines 212-215. The authors state that the invariant residue W558 serves a function of stabilizing the overall DBL structure through interactions with another tryptophan (W776). Why is the mutant KKKAIACK still folded? The invariant sequence PPR is also in this region. Is there any experimental evidence, such as mutagenesis, supporting your statement? Does W558 have interactions with other residues in this region, such as the adjacent helix (rightmost in Figure 2C)?

Mapping CS-protected sites - FPOP-MS. This is a potentially valuable set of experiments. However, the authors do not provide sufficient details on the experiments sensitivity coverage and only do technical replicates. The latter calls into question the use of a p value to make any conclusions. In fact, the p-

value is probably not particularly useful in any way to determine validity of the results. A more fruitful approach may be to a) provide sufficient information either as a supplemental or in the main-body of the paper to convince the reader of the quality of the MS data, and b) discuss their data in terms of cryoEM structure published recently. The authors should particularly focus on the issue of one or two sites for carbohydrate binding.

There are several improvements to figures that would better illustrate the points being made in the text.

Figure 1. (d) shows the semi-transparent EM density of the 3.8 Å structure with modeled coordinates in ribbon representation. DBL1 and ID1 are both red which makes it difficult to identify the region of the structure corresponding to the minimal binding region and is inconsistent with the rest of the figure which colors each domain uniquely – ID4 should also be distinct from DBL5. An additional panel showing the density/ribbon of the minimal binding region in the same orientation as (d) is recommended.

(f) shows a molecular surface of the model colored by charge – this is misleading – show either the EM density or the molecular surface for both d and f

Fig 3a. Statistical significance is not biological significance. The data needs to be deposited in an appropriate database. The results need to be discussed in more detail

Fig 4. Please ensure a and b are in identical orientations

Model Building and refinement. This section is unnecessarily vague. Please state clearly what each final model deposited in PDB contains.

Minor points

Fig S1 a should read levels not level.

Reviewer #2:

Remarks to the Author:

Summary:

Wang & Dagil, et al describe the structure biophysical characterization of the PfEMP1 VAR2CSA, which enables Plasmodium falciparum infected erythrocytes to attach to the placental tissue and avoid removal via splenic filtering. The authors find VAR2CSA forms a more compact conformation when in the presence of buffers with KCl and propose its importance in CS binding. They determine and report the structure of apo VAR2CSA and VAR2CSA in the presence of placental CS. Further assessment of the protein with and without the presence of CS by fast photochemical oxidation of proteins coupled with mass spectrometry supports the proposed CS binding site. The authors further explore the role of the WIW-motif and find it is crucial for CS-binding by exploring the similarities of WT and mutant DBL1-ID2 proteins by flow cytometry and Attana kinetics studies. Molecular docking and MD simulations further support the hypothesized CS-binding groove.

The data presented by the authors provide compelling evidence supporting the site of a putative CS-binding groove. Further, their structural determination offers new insight to the overall architecture of the protein. However, without direct visualization of the ligand (likely owing to the ligand's inherent heterogeneity or a more general heterogeneous binding) by cryo-EM, it would be worthwhile to assess the interaction with additional experiments in addition to their FPOP study before being acceptable for publication in Nature Communications. Additional points below would also further improve the manuscript.

Major points:

1. It is very striking that there is no density for pICS in the pICS reconstruction. It is reasonable to assume the extensive flexibility of the ligand is the cause, despite the fact the interaction is extremely high affinity. Nonetheless, lower resolution and low-occupancy features are often lost in high resolution reconstructions, as a consequence of high-resolution low-pass filtering (3.1Å in this case),

and aggressive B-factor sharpening; B-factors are presumably high for these data given the significant structural variability across the map. Did the authors examine maps with a lower resolution (e.g. 5-6Å) low-pass filter, with or without B-factor sharpening? This can restore low-resolution features, and may allow visualization of some pICS density in the proposed site, or elsewhere.

2. In Figure S1D, the authors show two maps “calculated using separate sub-fractions of particles.” This is also stated in the main text (Lines 172-175). Does this result from heterogenous refinement in cryoSPARC2? The authors should describe more precisely what led to the identification of these two classes. Further, this should be consistent with the procedure for the apo data, since the authors claim that this conformational change was only seen in the pICS dataset and is likely a result of pICS binding.

3. The authors claim that there are no structural changes in the core of VAR2CSA as a result of pICS binding. There should be a figure showing a superposition of the two structures, with C α RMSD values. This would also allow analysis of any effects on the DBL5/DBL6 domains, which are claimed to be different in the two data sets.

4. It would be helpful to show the side chains in Fig. 1E to assess potential interactions between ID3 and ID2, especially as Fig. S2 suggests the density in this area is of adequate resolution.

5. The suggestion that CS stabilizes the VAR2CSA core is unsubstantiated, especially because the authors point out that the structure of the DBL2-ID2-DBL3-DBL4-ID3 domains are remarkably similar. There are numerous variables (detergent, ice thickness, etc.) that could affect the increase in resolution seen in the core beyond presence of CS.

6. Claims of antigenic variability in the VAR2CSA DBL2 region would be better supported with an antibody binding assessment and/or kinetics experiments. Do antibodies that bind to this region in other PfEMP1s also bind to VAR2CSA DBL2? Do these antibodies bind with different affinities owing to the different groove entry point?

7. Pertaining to line 270: It is difficult to tell if greater flexibility is demonstrated in the DBL1 and DBL5/6 regions by cryo-EM without showing a heterogeneous refinement of the apo structure (Fig. S6 compared to Fig. S7).

8. Performing local refinements on the structures may help improve the resolution of the DBL1 and DBL5/6 regions. This can be done in cryosparc, though Relion tends to provide better results when dealing with flexible samples. It may be worthwhile extending the structural analysis and assessing whether local refinements and/or particle subtraction may increase the resolution in these areas and provide further insight as to whether this is CS-mediated or an artifact.

9. In Fig. S1B, it appears as though the local resolution legend is incorrect based on the reported resolution in the main text.

10. Is the resolution sufficient enough to notice changes in side chain orientation upon CS-binding in the proposed CS-binding groove?

Minor points:

1. Line 61: “...expressed on the surface of an infected erythrocytes at any given time”.

2. Line 79: “It is comprised composed of a short...”

3. Line 85: “...is comprised composed of...”

4. Line 92: “...recognized by VAR2CSA, is not only ...”

5. Line 105: “... induces high levels of inhibiting ...”

6. Line 112 and 113: “The structure supports shows that DBL2 is...”

7. Line 113: “...for charge-complementation for CS binding ...”

8. Line 177: “indicate that pICS do does not induce ...”

9. Line 198: “...DBL domains are comprised of comprise three subdomains ...”

10. Line 205: “the C-terminal domain of HB1 ...”

11. Line 207: “located in the C-terminal domain ...”

12. Line 208: “... variable region near this site, may reflect

13. Figure 3A: missing X and Y axes

14. Line 526: “The tip distal nitrogen atom of Lys/Arg residues side chains ...”

15. Line 558: “Local Rresolution and structural flexibility of VAR2CSA cryo-EM structure determined the cryoEM structure ”

16. Line 559: "... different levels... "
17. Line 561: "and shown at a ..."
18. Line 565: "displaced are DBL5/6" \diamond "DBL5/6 are displaced"
19. In line 84: CS has been defined in a previous paragraph (line 70)
20. In line 153: the phrase "directly linked" implies a covalent interaction, though this does not appear to be the case.
21. In line 214: labeling of W776 does not match Fig 2 (W766).
22. In line 350: This first sentence seems too colloquial.
23. Fig. S1A is missing the second local resolution legend.
24. While interesting, the discussion of this work's application to cancer therapy seems underdeveloped given the main focus on placental malaria.

Reviewer #3:

Remarks to the Author:

Wang et al reported the structure of a key placental malaria protein VAR2CSA. They used cryoEM to image VAR2CSA under two different conditions: without and with its binding ligand chondroitin sulfate A (CS). Even though CS can not be directly visualized in the solved structure, they used computational tool and biochemical approaches to identify interaction sites. Overall the results are novel and important. The manuscript should be accepted for publication after the following points are addressed.

Major points:

- 1, In the cryoem data collection details, the apo sample was collected with the camera operated at linear mode (line 329), but the CS treated sample was collected with the camera operated in the counted mode (line 342). Coincidentally the reported resolution for the apo structure is significantly lower than the CS treated sample. This makes one wonder if the apo structure resolution could have been improved should the camera be operated in the counted mode and why they did not collect both data sets in the counted mode.
- 2, In most of the FSC curves presented for both structures, there are a slow and wide dip around 6 angstroms. Can the authors provide an explanation to this dip?
- 3, In the "imbalance" hetero refinement, which tool did the author use for low pass filtering on the map (line 352)? Some tools do low pass filtering to a certain resolution by suppressing the high-resolution Fourier amplitudes, some tools do this by performing phase randomization, some include both. The phase randomization type should be required here according to the "random-phase 3D classification" method (line 351, reference 34).
- 4, Since there are many parameters (e.g. low pass filter resolution, initial resolution, box size, number of iterations, number of "trash can" models) to adjust in the "imbalance" hetero refinement (line 357-358), the author should provide a supplemental table to show the adjustment of these parameters at each step, the number of remaining particles, the number of removed particles, map resolution and so on.
- 5, The phase randomization was introduced in each iteration of 3D classification in the paper of "random-phase 3D classification" (Gong et al, 2016, reference 34), but it was only introduced once at the beginning of each heterogeneous refinement in this manuscript. Could the resolution be further improved if the original "random-phase 3D classification" method is applied?
- 6, In Fig. S6c, it is shown that "UN-refinement. Apo structure as initial model. 5 classes" after 2D classification in that workflow. Should it be "Heterogeneous refinement" (which generates multiple maps), not "UN-refinement" (which generates one map) in Cryosparc? What is the reason to use apo structure as initial model, but not build the initial model from the CS treated dataset itself?

Minor points:

1, In line 328 and line 342, the author mentioned "Falcon K3 direct detector camera". Falcon and K3 are two different direct electron detectors from two different companies. Does the author mean "Falcon 3 direct electron detector"?

2, In line 351, "by Xin et. al." should be "by Gong et al." when citing the reference 34.

3, In line 352, the final map is low-pass-filter to resolution 20A. However, Fig. S5c shows that the "Trash can" input model was low-pass-filtered to 40A. Please make them consistent.

4, In Fig.S6c, 5 classes are shown before UN-refinement. Were class 4 (28.78%) un-selected for the next UN-refinement? The number of particles is 627,711 after 2D classification. The number of particles in UN-refinement is 456,192 before the "imbalance" hetero refinement. It seems like 27.32% particles were removed before UN-refinement, but no class corresponds to the percentage 27.32% in these 5 classes.

Reviewer #4:

Remarks to the Author:

The manuscript reports a Cryo-EM structure of VAR2CSA, an important protein found in the parasites that cause placental malaria. Despite structural understanding of several segments within VAR2CSA, the overall structure of VAR2CSA ectodomain is previously not well understood. In this manuscript, the authors report a Cryo-EM structure of VAR2CSA ectodomain and identified a positively charged groove on the surface that is responsible for partner binding. Through fast photochemical oxidation of proteins (FPOP) coupled with mass spectrometry (MS) analysis, the binding interaction between VAR2CSA ectodomain and the ligand Chondroitin Sulfate A (CS) is characterized in solution. Such interaction is then visualized by molecular docking. As the ectodomain is relatively conserved among parasites, such understanding of VAR2CSA structure provides valuable insights in helping the future development of vaccines against placental malaria. Overall, the manuscript is well presented, and the conclusions are supported by experimental data. The topic is of broad interest and fits in the scope of Nature Communications and can be considered for publication after addressing these concerns listed below.

1. Line 84, authors used CS as abbreviation of Chondroitin Sulfate A. Later in the manuscript, authors used both CS and CSA (e.g., lines 101, 104, 247, etc.) to refer Chondroitin Sulfate A. Authors should keep the abbreviation consistent and go through the manuscript to double-check the use of abbreviations.

2. Line 115 and line 248, authors mentioned that an additional tryptophan residue is unusual for DBL domains. It will be helpful if authors can provide more context and expand a little bit on the current understanding of a typical DBL domain.

3. Lines 136-138, authors mentioned that the binding affinities in NaCl or KCl solutions differ by 10-fold. Such conclusion is from fitting of the QCM biosensor interaction data. The fitting of the binding under two electrolyte conditions were by two different fitting models (1:1 for KCl and 1:2 for NaCl). Can authors comment on why two fitting models were used rather than using the same model to fit both curves?

4. Lines 153-155, authors mentioned that ID2 and ID3 are directly linked through the open-mouth shape conformation of the ID2 C-terminus. The authors also visualized such interaction through Figures 1e, S2a, and S2b. However, such visualization is not easy to follow. It will be helpful if the

authors can mark the residues numbers in Figure 1e and refer to Figure S2a and S2b for high-res structure. Or authors can insert an overall structural model of this region in Figure S2 for a better presentation.

5. Lines 167-194, authors presented the cryo-EM observation of the VAR2CSA-pICS complex and found that the core region is stabilized by pICS binding but the peripheral region is more flexible. Since the pICS is highly flexible and is challenging to be resolved in Cryo-EM, is there other evidence supporting the formation of VAR2CSA-pICS complex?

6. Line 234, authors claim that FPOP-MS identifies potential binding regions primarily in ID1, DBL2, and ID2 by referring to Figures 3a and S8. It is challenging to link the peptides to the structure based on the current presentation. I would recommend a coverage map that connects the sequence to various structural domains in supporting information, or at least label the regions/peptides that show protections upon binding with pICS with residues numbers in Figure S8.

7. In general, FPOP-MS or other covalent labeling approaches are capable to identify critical binding residues owing to the irreversible nature of the labeling. I wonder if the authors identified any critical binding residues in this study. If so, it will greatly elevate the spatial resolution of the binding identification and such information should be reported. If not, the authors should comment on why only peptide-level data is available and why residue-level resolution cannot be achieved in this study.

8. To follow up with previous comment, it will be helpful if the authors can present a representative extracted ion chromatogram for the wildtype and the hydroxyl radical modified species with corresponding MS/MS spectrum to justify their argument.

9. Can the authors comment on the sequence coverage of the FPOP-MS analysis? If the sequence coverage is not (near) 100%, how did the authors rule out the possibilities that other regions that are not covered may also involve in binding?

10. Lines 254-256, authors compared the melting curves for both wildtype and the mutant protein and emphasized that a clear transition from folded to unfolded was observed for the wildtype protein. Correct me if I'm not understanding it properly, but a clear transition is observed for the mutant as well (Figure 3c). If so, why is the "clear transition for wildtype" worth emphasizing?

11. Lines 264-266, authors mentioned that the docking model was obtained based on all structural and biochemical results. Authors also mentioned the experimental procedures of the docking in lines 388-392. Can authors expand the discussion on how the structural and biochemical results were utilized in docking? Specifically, how FPOP-MS data contribute to the docking process?

12. Lines 435-436, 1 mM adenine was mentioned twice.

13. It seems that the pICS is a hydroxyl radical scavenger and the advantage of using adenine as a dosimeter is to make sure the effective radical concentration is comparable under both pICS-free and pICS-bound states. It will be helpful if the authors can show some evidence (e.g., dosimetry readings) demonstrating the comparable effective radical concentration is achieved, so that the comparison in Figure 3a is better justified.

Point to point reply to review comments:

Report from Reviewer 1:

This manuscript describes the structures of VAR2CSA from FCR3 in the presence and absence of CS. It builds upon a lower resolution structure of the 3D7 variant published by others (BioRxiv April 2020; JBC Oct 2020) that identified the relative locations of DBL domains that created two pores involving DBL1-ID1-DBL2-ID2-DBL4. This earlier paper proposed a model for how VAR2CSA evaded immune detection and predicted the two CS binding sites, identified as high and low affinity in the current manuscript, and should be fully attributed for these contributions. For example, the structures described in this manuscript provide more details and support the previously published model that identified the relative location of domains and two pores/grooves as potential CS binding regions. This is contrary to your statement on lines 97-99.

Reply: Thank you for raising this point. We have modified the manuscript to include a discussion on the binding site published in JBC April 2020).

A second more recent publication describing the structures of FCR and N strains by cryoEM has been published and all data is available (deposited in July 2020 and released Jan 13th 2021). Although now published, it occurred after this manuscript had been submitted and therefore a note to this effect needs to be added. However, the work described here is largely confirmatory and the two publications set the bar higher for the current manuscript. The data in this manuscript were deposited into PDB in Dec 2020 (7B52 and 7B54) and a summary table of data collection, refinement statistics included. Although this demonstrates that your results were finalized later, due to the closeness in time, I have not taken the results of this second paper into account when reviewing your manuscript. You may however, wish to incorporate the results more fully into your discussion as there are some interesting differences.

Reply: Yes thank you we are now aware of these data and have included a discussion and reference to this work both in the introduction and in the result section.

In general, the contributions of groups outside of your own have been, at best, overlooked in favor of self-citation. For example, the SAXS envelope described by Srivastava in a 2010 PNAS publication provided the first view of the ectodomain whereas the later work cited did not correctly assign relative positions ID and DBL domains described here. No one works in a vacuum and it is better to cite significant contributions appropriately. This needs to be fully addressed in the current manuscript.

Reply: Yes we agree, and point taken. We have modified the introduction accordingly.

The data processing and model refinement statistics are poorer than I would expect. The percentage of residues in favoured areas of the Ramachandran plot seems low especially considering Ramachandran restraints were imposed (line 370) and the rmsd in bond angles and lengths too broad suggesting geometric restraints were too loose. How do you account for this? Can this be improved?

Reply: The poor structure refinement statistics are mainly due to the local flexibility of the DBL5/6 tail and part of DBL1. The reported resolution is calculated mainly based on the core structure from DBL2 to DBL4. For the pICS complex dataset, the overall resolution of the map is calculated to be 3.1Å, the center of the core structure could even reach about 2.8Å. However, the DBL5/6 tail has a resolution of 6-10Å and the refinement for the entire structure has poor statistics due to the contribution of DBL5/6. In the revised manuscript we have further processed the VAR2CSA:pICS dataset and the local resolution of DBL5/6 was improved to around 4 Å. Due to this, we have submitted an additional pdb file that only cover the DBL5/6 region. The data processing and structure refinement details are updated in the methods section and in the supplementary figures.

The protein produced by baculovirus is glycosylated. How do you think this impacts access to the CS binding surfaces? Could it have an effect?

Reply: Yes that could have an effect. However, in our previous work with this full length VAR2CSA protein produced in insect cells we have examined the potential effect of glycosylation on CSA binding by either producing the protein in the presence of tunicamycin (which results in a non-glycosylated protein) as well as rigorously deglycosylation. At least for the insect cell proteins it does not seem that glycosylation effects affinity to CSA.

It is surprising that CS-bound VAR2CSA increases the lability of DBL1. The description of interfacial residues involving DBL1 is vague and needs to be described in more detail. It would be helpful to color domains individually to

better understand the relative contributions to the CS binding site (for example in figure 4b) and individual amino acids in the groove (figure 4d). This information is presented in linear form in Fig S7, so will be straightforward to apply to the structure.

Reply: We thank the reviewer for the suggestion. Given the lability of DBL1 in the complex of VAR2:pICS we cannot give a detailed description of the residues involved in binding from this domain, as it would be based on the docking model and not the cryoEM structure. The CS binding groove is shown in figure 2b with the individual amino acid residues labelled. We have updated figure 1d, which now shows front/back view of the full structure with all parts coloured individually to match figure S7, also we updated figure 4b. Residues that partake in the CS binding according to the docking studies are now indicated in updated Fig. 4d.

In figure 1c, it is intriguing that binding measurements in the presence of 75 mM KCl results in 1:1 binding stoichiometry whereas in the presence of 75 mM NaCl, the stoichiometry is 1:2 since two pores were observed in the EM structures (neg stain and here) and earlier work by you and others show 1:1 stoichiometry in NaCl. Please account for this?

Reply: Yes, this is an interesting observation. The stoichiometry of binding in 75 mM NaCl was fitted using 1:2 model, as the dissociation constant could not be fitted using a 1:1 model in this case. It is important to point out that the comparison is shown to highlight the affinity in KCl is 10x higher compared to NaCl. The same QCM sensor chip was used for both samples, however, whether or not the second minor groove proposed binding site observed in the published EM structures is accessible in NaCl and not KCl, the QCM experiment cannot confirm, but it cannot be excluded either. We have updated the manuscript to address this point.

Crystal structures of DBL domains in complex with their cellular receptors identify similar regions of the DBL fold for receptor binding. Do the structures described in this manuscript explain why VAR2CSA does not bind these receptors?

Reply: DBL domains all form the same basic scaffold, which is upheld by core helical elements. All of the conserved residues of these core elements are buried deep in the fold. The core elements of different DBL domain thus have homologous functions; they are not identical in sequence but they can be recognized in sequence by their shared sequence motifs, i.e. by the described homology blocks. Variations in the loops, elongated and additional helices decorating the DBL domains account for the different binding specificities, as does the domain packing into dimers or complex macro-molecular structures as in the case of e.g. the glycan-binding EBA175 and VAR2CSA proteins. So while the DBL fold present binding regions at similar positions in the domains, these regions themselves differ in structure and chemical composition. This resemblance and diversity between VAR2CSA and general DBL folds has been reformulated to better convey these notions.

Lines 212-215. The authors state that the invariant residue W558 serves a function of stabilizing the overall DBL structure through interactions with another tryptophan (W776). Why is the mutant KKKAIACK still folded? The invariant sequence PPR is also in this region. Is there any experimental evidence, such as mutagenesis, supporting your statement? Does W558 have interactions with other residues in this region, such as the adjacent helix (rightmost in Figure 2C)?

Reply: In other DBL domains (e.g. EBA175, ICAM1-binding DBLs) the conserved Pro in PPR (position 4 in HB4) introduces a kink in upstream β -sheet structures of S1 (no beta sheets are present in DBL2 VAR2CSA), and is in a position to interact hydrophobically with the also conserved Trp on position 8 in HB2 (corresponding to W766) as well as the conserved W in corresponding to position W558. In VAR2CSA DBL2 the organization of this triplet is different. Instead of organized as sequence appearance N-term W-P-W C-term, as in other DBLs, the VAR2CSA DBL2 has residues ordered as W558 –W766-Pro. A number of other Try –Try interactions stabilize the DBL S2 and S3 fold. So it is likely that the KKKAIACK mutation has caused a local structural change not affecting the basic DBL2 S2 and S3 fold, or the overall macro-molecular fold. We have modified the manuscript accordingly, to include this point.

Mapping CS-protected sites - FPOP-MS. This is a potentially valuable set of experiments. However, the authors do not provide sufficient details on the experiments sensitivity coverage and only do technical replicates. The latter calls into question the use of a p value to make any conclusions. In fact, the p-value is probably not particularly useful in any way to determine validity of the results. A more fruitful approach may be to a) provide sufficient information either as a supplemental or in the main-body of the paper to convince the reader of the quality of the MS data, and b) discuss their data in terms of cryoEM structure published recently. The authors should particularly focus on the issue of one or two sites for carbohydrate binding.

Reply: We believe that the technical replicates yield a usable p-value. The technical replicates were from separate FPOP labelling experiments of aliquots from a single protein (and, where applicable, placental CS) preparation. The p-value captures the variability in labelling, quenching, proteolysis, and LC-MS analysis; all of these factors are known to be able to result in considerable variability in FPOP results (see DOI:10.1007/s13361-018-1994-y for more discussion). Any problems stemming from poor quality MS data will be captured in these replicates. What these replicates do not capture are any topographical differences between protein preparations, or binding differences in CS preparations. However, we generally agree with the reviewer that the community is better served by transparent presentation of the data when possible. We have made the raw LC-MS/MS data available on the ProteomeXchange server and will release it publicly upon acceptance of the manuscript. The accession number is PXD024154; the reviewer login is reviewer_pxd024154@ebi.ac.uk and the password is YUw37Znb. We have also added selected Extracted Ion Chromatograms (EICs) to the Supplementary Information S9.

There are several improvements to figures that would better illustrate the points being made in the text. Figure 1. (d) shows the semi-transparent EM density of the 3.8 Å structure with modelled coordinates in ribbon representation. DBL1 and ID1 are both red which makes it difficult to identify the region of the structure corresponding to the minimal binding region and is inconsistent with the rest of the figure which colours each domain uniquely – ID4 should also be distinct from DBL5. An additional panel showing the density/ribbon of the minimal binding region in the same orientation as (d) is recommended.(f) shows a molecular surface of the model colored by charge – this is misleading – show either the EM density or the molecular surface for both d and f

Reply: Yes, thank you for the suggestions. We have updated figure 1d, to highlight the ID1 in another colour and have added a backside view of the full VAR2CSA structure to show its position. This should make it clear to the reader where the boundaries between domains are located in the structure. We have removed panel f in figure 1, and instead added a molecular dynamics movie as supplementary material, showing surface charge of VAR2CSA calculated using a series of MD simulated models.

Fig 3a. Statistical significance is not biological significance. The data needs to be deposited in an appropriate database. The results need to be discussed in more detail

Reply: Please see comment above.

Fig 4. Please ensure a and b are in identical orientations

Reply: Please see comment above.

Model Building and refinement. This section is unnecessarily vague. Please state clearly what each final model deposited in PDB contains.

Reply: We have updated the Table for refinement and added details about submitted pdb files.

Minor points

Fig S1 a should read levels not level.

Reply: Figure S1 has been updated, as the structural refinement was further processed during the re-submission process. The figure legend has been updated as well.

Report from Reviewer 2:

Summary:

Wang & Dagil, et al describe the structure biophysical characterization of the PfEMP1 VAR2CSA, which enables Plasmodium falciparum infected erythrocytes to attach to the placental tissue and avoid removal via splenic filtering. The authors find VAR2CSA forms a more compact conformation when in the presence of buffers with KCl and propose its importance in CS binding. They determine and report the structure of apo VAR2CSA and VAR2CSA in the presence of placental CS. Further assessment of the protein with and without the presence of CS by fast photochemical oxidation of proteins coupled with mass spectrometry supports the proposed CS binding site. The authors further explore the role of the WIW-motif and find it is crucial for CS-binding by exploring the similarities of WT and mutant DBL1-ID2 proteins by flow cytometry and Attana kinetics studies. Molecular docking and MD simulations further support the hypothesized CS-binding groove.

The data presented by the authors provide compelling evidence supporting the site of a putative CS-binding groove. Further, their structural determination offers new insight to the overall architecture of the protein. However, without direct visualization of the ligand (likely owing to the ligand's inherent heterogeneity or a more general heterogeneous binding) by cryo-EM, it would be worthwhile to assess the interaction with additional experiments in addition to their

FPOP study before being acceptable for publication in Nature Communications. Additional points below would also further improve the manuscript.

Thank you for your comments and suggestions. Since the submission of our manuscript another article (Ma et al., 2021) has been published which confirms our proposed binding region, we have referenced and discussed these data in context. Please see our point to point comments below:

Major points:

1. It is very striking that there is no density for pICS in the pICS reconstruction. It is reasonable to assume the extensive flexibility of the ligand is the cause, despite the fact the interaction is extremely high affinity. Nonetheless, lower resolution and low-occupancy features are often lost in high resolution reconstructions, as a consequence of high-resolution low-pass filtering (3.1Å in this case), and aggressive B-factor sharpening; B-factors are presumably high for these data given the significant structural variability across the map. Did the authors examine maps with a lower resolution (e.g. 5-6Å) low-pass filter, with or without B-factor sharpening? This can restore low-resolution features, and may allow visualization of some pICS density in the proposed site, or elsewhere.

Reply: Thank you for the comments and suggestions. We have done further processing of the datasets including low-pass filtering of the maps without sharpening and we could not identify densities that could represent ligand. We think the main reason this is that our sample is purified from its natural source and therefore is heterogeneous, as the reviewer points out. To prepare the sample for cryoEM, we did gel filtration after treating the sample with chondroitinase ABC to remove unprotected regions, but in our ligand binding model, part of the ligand is still surface accessible, perhaps contributing to why no continuous ligand density was detected. There are however indirect evidence of complex formation, for example the gel filtration profile changed after complex formation, and also DBL1 is less well-resolved in the pICS sample compared to the apo sample.

2. In Figure S1D, the authors show two maps “calculated using separate sub-fractions of particles.” This is also stated in the main text (Lines 172-175). Does this result from heterogeneous refinement in cryoSPARC2? The authors should describe more precisely what led to the identification of these two classes. Further, this should be consistent with the procedure for the apo data, since the authors claim that this conformational change was only seen in the pICS dataset and is likely a result of pICS binding.

Reply: The two maps were calculated using heterogeneous refinement as shown in Fig. S6, using three identical starting model and we have also updated the figure S1 to make this more clear. We have processed the apo data in a similar manner to show the differences of the two datasets. It is evident that the pICS dataset has more heterogeneity.

3. The authors claim that there are no structural changes in the core of VAR2CSA as a result of pICS binding. There should be a figure showing a superposition of the two structures, with Ca RMSD values. This would also allow analysis of any effects on the DBL5/DBL6 domains, which are claimed to be different in the two data sets.

Reply: From the pICS dataset, we generated two maps that are structurally identical in the core region, but with different DBL5/6 orientation. In the new Fig. S1, we have revised the alignment of the two structures together and the with RMSD of the structures is noted in the figure legend.

4. It would be helpful to show the side chains in Fig. 1E to assess potential interactions between ID3 and ID2, especially as Fig. S2 suggests the density in this area is of adequate resolution.

Reply: We thank the reviewer for this suggestion, we have updated Fig 1e to show the side chain interaction between ID3 and ID2.

5. The suggestion that CS stabilizes the VAR2CSA core is unsubstantiated, especially because the authors point out that the structure of the DBL2-ID2-DBL3-DBL4-ID3 domains are remarkably similar. There are numerous variables (detergent, ice thickness, etc.) that could affect the increase in resolution seen in the core beyond presence of CS.

Reply: We agree that there are difference between the two prepared samples. The reason we claim that CS stabilizes the VAR2CSA is not because the pICS sample demonstrates higher resolution overall, but rather because the DBL1 and DBL5/6 are relatively less well-resolved compared to the core structure. In contrast, although the apo data has an overall lower resolution, the DBL1 density is in fact much better compared to the pICS dataset. We have modified Fig S1 to include a comparison of the map resolution of DBL1, the core region and DBL5/6 separately to show the relative flexibility of the datasets. Moreover, we have toned down the claim in the manuscript.

6. Claims of antigenic variability in the VAR2CSA DBL2 region would be better supported with an antibody binding assessment and/or kinetics experiments. Do antibodies that bind to this region in other PfEMP1s also bind to VAR2CSA DBL2? Do these antibodies bind with different affinities owing to the different groove entry point?

Reply: While the proposed CSA binding site of DBL2 is homologous in function and location (glycan binding site) in EBA175, the CSA binding loop is diverse in sequence as well as sequence length among var2csa variants, and thus likely a diversification of importance for antibody recognition. However, there is very limited sequence and structural conservation to other PfEMP1, and therefore it is very unlikely that antibodies would bind across VAR2CSA and other types of PfEMP1. There are extensive publications testing sera from highly immune children against placental malaria parasites demonstrating limited or no cross-reactivity, and no monoclonal antibodies to this regions have been identified. However, we are very excited to see how we can explore this and recent structural information to target said region for a next generation VAR2CSA placental malaria vaccine.

7. Pertaining to line 270: It is difficult to tell if greater flexibility is demonstrated in the DBL1 and DBL5/6 regions by cryo-EM without showing a heterogeneous refinement of the apo structure (Fig. S6 compared to Fig. S7).

The main reason why the two dataset is not processing in the identical way is that the two samples are prepared in different ways. The apo sample is prone to aggregate during freezing and we have used FF8 as additive.

Reply: While FF8 sharpens the contrast of the particles, the particle concentration is becoming lower, as also shown in Supplementary S5 and S6. In average, there are only about 50 particles per image with FF8, so we have collected in total 8000+ images for this sample, yielding only about 300K particles following 2D classification.

In the revised manuscript we have processed the apo dataset using a similar strategy as for the pICS data and updated Fig. S5 accordingly.

8. Performing local refinements on the structures may help improve the resolution of the DBL1 and DBL5/6 regions. This can be done in cryosparc, though Relion tends to provide better results when dealing with flexible samples. It may be worthwhile extending the structural analysis and assessing whether local refinements and/or particle subtraction may increase the resolution in these areas and provide further insight as to whether this is CS-mediated or an artifact.

Reply: We thank the reviewer for this suggestion, also received from reviewer 1. Please see comment to this in R1 comment, and the associated new pdb file covering DBL5/6 portion.

9. In Fig. S1B, it appears as though the local resolution legend is incorrect based on the reported resolution in the main text.

Reply: We have now revised Fig. S1 and its legend.

10. Is the resolution sufficient enough to notice changes in side chain orientation upon CS-binding in the proposed CS-binding groove?

Reply: The resolution is highest in the centre of the domains while the side chains that are present on the surface of the structure are typically usually less well resolved. Hence it is unlikely that sidechain orientations can be fully identified.

Minor points:

1. Line 61: "...expressed on the surface of an infected erythrocytes at any given time".
2. Line 79: "It is comprised composed of a short..."
3. Line 85: "...is comprised composed of..."
4. Line 92: "...recognized by VAR2CSA, is not only ..."
5. Line 105: "... induces high levels of inhibiting ..."
6. Line 112 and 113: "The structure supports shows that DBL2 is..."
7. Line 113: "...for charge-complementation for CS binding ..."
8. Line 177: "indicate that pICS do does not induce ..."
9. Line 198: "...DBL domains are comprised of comprise three subdomains ..."
10. Line 205: "the C-terminal domain of HB1 ..."
11. Line 207: "located in the C-terminal domain ..."
12. Line 208: "... variable region near this site, may reflect
13. Figure 3A: missing X and Y axes –
14. Line 526: "The tip distal nitrogen atom of Lys/Arg residues side chains ..."
15. Line 558: "Local Rresolution and structural flexibility of VAR2CSA cryo-EM structure determined the cryoEM structure"
16. Line 559: "... different levels..."
17. Line 561: "and shown at a ..."
18. Line 565: "displaced are DBL5/6" à "DBL5/6 are displaced"
19. In line 84: CS has been defined in a previous paragraph (line 70)
20. In line 153: the phrase "directly linked" implies a covalent interaction, though this does not appear to be the case.

21. In line 214: labeling of W776 does not match Fig 2 (W766).
22. In line 350: This first sentence seems too colloquial.
23. Fig. S1A is missing the second local resolution legend.
24. While interesting, the discussion of this work's application to cancer therapy seems underdeveloped given the main focus on placental malaria.

Reply: All minor points have been updated in the manuscript. Regarding comment 13 and 23. Figure 3A and figure S1 have both been updated.

Report from Reviewer 3:

Wang et al reported the structure of a key placental malaria protein VAR2CSA. They used cryoEM to image VAR2CSA under two different conditions: without and with its binding ligand chondroitin sulfate A (CS). Even though CS can not be directly visualized in the solved structure, they used computational tool and biochemical approaches to identify interaction sites. Overall the results are novel and important. The manuscript should be accepted for publication after the following points are addressed.

We thank the reviewer for the positive comments, please see below for our point-to-point replies.

Major points:

1, In the cryoem data collection details, the apo sample was collected with the camera operated at linear mode (line 329), but the CS treated sample was collected with the camera operated in the counted mode (line 342). Coincidentally the reported resolution for the apo structure is significantly lower than the CS treated sample. This makes one wonder if the apo structure resolution could have been improved should the camera be operated in the counted mode and why they did not collect both data sets in the counted mode.

Reply: The apo sample is optimized with adding Fluorinated-foscholin-8 (FF8). While FF8 improves the contrast of the particles, the particle concentration is much lower, as shown in supplementary figures S5 and S6. In average, there are only about 50 particles per image in the apo data, so we have collected in total 8000+ images for this sample. The microscope in our facility only has Falcon3 camera, which is relatively slow in counting mode (before EPU upgrade, around 30 images were retrieved per hour when we collect the dataset), so we have chosen to collect the data in linear mode. We anticipate that collecting the same dataset size in counting mode will improve the final resolution, but it will not change the relative flexibility, and we would not gain much more information compared to current dataset.

2, In most of the FSC curves presented for both structures, there are a slow and wide dip around 6 angstroms. Can the authors provide an explanation to this dip?

Reply: The sharp dip in the corrected FSC curves shows where phase randomization begins in the calculation of corrected FSC. In our experience, the slow dip at around 6Å is quite common in cryosparc refinement, as discussed in cryosparc fora such as <https://discuss.cryosparc.com/t/extra-bump-in-fsc/2170>. For this particular project, we think it is caused by the flexibility of DBL1/5/6 region. With our data processing method, the final dataset used to calculate the map consists of at least two sub-datasets, both with identical core structure, but with DBL5/6 placed differently. The heterogeneity of the final dataset renders the correlation of the map lower at low resolution (about 6Å, which could represent domain movement). Yet, the high-resolution signal contributed by the core structure is higher than further splitting of the datasets.

3, In the “imbalance” hetero refinement, which tool did the author use for low pass filtering on the map (line 352)? Some tools do low pass filtering to a certain resolution by suppressing the high-resolution Fourier amplitudes, some tools do this by performing phase randomization, some include both. The phase randomization type should be required here according to the “random-phase 3D classification” method (line 351, reference 34).

Reply: All data processing was employed in cryosparc and low passing filtering was done using Cryosparc Volume tools and we have updated the methods section accordingly.

4, Since there are many parameters (e.g. low pass filter resolution, initial resolution, box size, number of iterations, number of “trash can” models) to adjust in the “imbalance” hetero refinement (line 357-358), the author should provide a supplemental table to show the adjustment of these parameters at each step, the number of remaining particles, the number of removed particles, map resolution and so on.

Reply: We have included an additional table (Table S2) to illustrate the process of the “imbalance” hetero refinement to show the trend of percentage of “poor particles” removed in each iteration with different parameters, and to also demonstrate how much the resolution was improved in each iteration.

5, The phase randomization was introduced in each iteration of 3D classification in the paper of “random-phase 3D

classification” (Gong et al, 2016, reference 34), but it was only introduced once at the beginning of each heterogeneous refinement in this manuscript. Could the resolution be further improved if the original “random-phase 3D classification” method is applied?

Reply: The “imbalanced” hetero refinement that we present in this manuscript is not identical to the “random-phase 3D classification” method. We were inspired by the cited paper, but applied the principle differently. Our method is solely performed in cryosparc, without any need for scripting, which is convenient for non-experts. There may be others who have attempted similar approaches as discussed in this cryosparc forum (<https://discuss.cryosparc.com/t/removing-damaged-bad-particles-by-multireference-refinement-with-low-pass-filtered-references/480>), but to our knowledge no one has published it before. We do not know if the original “random-phase 3D classification” method would improve the outcome. However, in our hands the data processing in Relion did not perform as good as cryosparc for this particular project.

6, In Fig. S6c, it is shown that “UN-refinement. Apo structure as initial model. 5 classes” after 2D classification in that workflow. Should it be “Heterogeneous refinement” (which generates multiple maps), not “UN-refinement” (which generates one map) in Cryosparc?

Reply: It should be “Heterogeneous refinement” and we have corrected this, thank you.

What is the reason to use apo structure as initial model, but not build the initial model from the CS treated dataset itself?

Reply: The VAR2CSA shape with domains on the same plane, render the particles having very different projection views. Thus, the resolution of certain, rare 2D views is very low in the initial 2D classification and may hence be discarded as bad classes despite that there is useful information in them. For the apo structure, we have only used particles from five very high resolution 2D averages to generate the initial model and then used “imbalanced” heterogeneous refinement to discard bad particles step-by-step instead of generating several initial models and run classical heterogeneous refinement. Providing a good initial model was very important in this project to keep as many good particles as possible to reach high resolution, so we have used the apo structure as initial models in the CS treated dataset also.

Minor points:

1, In line 328 and line 342, the author mentioned “Falcon K3 direct detector camera”. Falcon and K3 are two different direct electron detectors from two different companies. Does the author mean “Falcon 3 direct electron detector”?

Reply: This has now been corrected, thanks for pointing it out.

2, In line 351, “by Xin et. al.” should be “by Gong et al.” when citing the reference 34.

Reply: We have corrected this.

3, In line 352, the final map is low-pass-filter to resolution 20A. However, Fig. S5c shows that the “Trash can” input model was low-pass-filtered to 40A. Please make them consistent.

Reply: We have corrected this in the manuscript and include more details in Table S2.

4, In Fig.S6c, 5 classes are shown before UN-refinement. Were class 4 (28.78%) un-selected for the next UN-refinement? The number of particles is 627,711 after 2D classification. The number of particles in UN-refinement is 456,192 before the “imbalance” hetero refinement. It seems like 27.32% particles were removed before UN-refinement, but no class corresponds to the percentage 27.32% in these 5 classes.

Reply: Class 1, 4, 5 were all used for next round UN-refinement. We have added arrows in the figure S6 to make this clearer. The three classes have identical core structures but different occupancy in DBL1/5/6 region. Inclusion of all three classes generated the highest resolution map of the core structure.

Reviewer #4 (Remarks to the Author):

The manuscript reports a Cryo-EM structure of VAR2CSA, an important protein found in the parasites that cause placental malaria. Despite structural understanding of several segments within VAR2CSA, the overall structure of VAR2CSA ectodomain is previously not well understood. In this manuscript, the authors report a Cryo-EM structure of VAR2CSA ectodomain and identified a positively charged groove on the surface that is responsible for partner binding. Through fast photochemical oxidation of proteins (FPOP) coupled with mass spectrometry (MS) analysis, the binding interaction between VAR2CSA ectodomain and the ligand Chondroitin Sulfate A (CS) is characterized in solution. Such interaction is then visualized by molecular docking. As the ectodomain is relatively conserved among parasites,

such understanding of VAR2CSA structure provides valuable insights in helping the future development of vaccines against placental malaria. Overall, the manuscript is well presented, and the conclusions are supported by experimental data. The topic is of broad interest and fits in the scope of Nature Communications and can be considered for publication after addressing these concerns listed below.

1. Line 84, authors used CS as abbreviation of Chondroitin Sulfate A. Later in the manuscript, authors used both CS and CSA (e.g., lines 101, 104, 247, etc.) to refer Chondroitin Sulfate A. Authors should keep the abbreviation consistent and go through the manuscript to double-check the use of abbreviations.

Reply: Thank you for highlighting this, we agree that a consistent abbreviation is preferred. This has been corrected throughout the manuscript.

2. Line 115 and line 248, authors mentioned that an additional tryptophan residue is unusual for DBL domains. It will be helpful if authors can provide more context and expand a little bit on the current understanding of a typical DBL domain.

Reply: The first introduction of the typical DBL fold has been expanded. And the description of the WIW motif has been altered to clarify the difference to most other DBL domains.

3. Lines 136-138, authors mentioned that the binding affinities in NaCl or KCl solutions differ by 10-fold. Such conclusion is from fitting of the QCM biosensor interaction data. The fitting of the binding under two electrolyte conditions were by two different fitting models (1:1 for KCl and 1:2 for NaCl). Can authors comment on why two fitting models were used rather than using the same model to fit both curves?

Reply: Please see answer to the same question above to Reviewer 1

4. Lines 153-155, authors mentioned that ID2 and ID3 are directly linked through the open-mouth shape conformation of the ID2 C-terminus. The authors also visualized such interaction through Figures 1e, S2a, and S2b. However, such visualization is not easy to follow. It will be helpful if the authors can mark the residues numbers in Figure 1e and refer to Figure S2a and S2b for high-res structure. Or authors can insert an overall structural model of this region in Figure S2 for a better presentation.

Reply: This clarification was also raised by Reviewer 1, figures have been updated accordingly.

5. Lines 167-194, authors presented the cryo-EM observation of the VAR2CSA-pICS complex and found that the core region is stabilized by pICS binding but the peripheral region is more flexible. Since the pICS is highly flexible and is challenging to be resolved in Cryo-EM, is there other evidence supporting the formation of VAR2CSA-pICS complex?

Reply: Following extensive purification of CS from placental tissue, yes there would indeed be the risk that it could not bind VAR2CSA. However this was thoroughly examined in several assays to ensure that we have purified a relevant CS moiety. We have now included a biosensor assay where VAR2CSA binding is measured to biotinylated pICS (see figure 1F) and the data show a high affinity interaction.

6. Line 234, authors claim that FPOP-MS identifies potential binding regions primarily in ID1, DBL2, and ID2 by referring to Figures 3a and S8. It is challenging to link the peptides to the structure based on the current presentation. I would recommend a coverage map that connects the sequence to various structural domains in supporting information, or at least label the regions/peptides that show protections upon binding with pICS with residues numbers in Figure S8.

Reply: We have updated Fig S7 and S8 to show plots of all sites of protection, exposure, and oxidized regions that are unchanged upon pICS binding.

7. In general, FPOP-MS or other covalent labeling approaches are capable to identify critical binding residues owing to the irreversible nature of the labeling. I wonder if the authors identified any critical binding residues in this study. If so, it will greatly elevate the spatial resolution of the binding identification and such information should be reported. If not, the authors should comment on why only peptide-level data is available and why residue-level resolution cannot be achieved in this study.

Reply: The accurate determination of residue-level FPOP data requires significantly more sample than peptide-level, especially for a protein as large as VAR2CSA. As we were sample-limited for this study, we were unable to perform the large number of runs and the large sample injections required to generate reliable residue-level data. However, we did go back and performed semi-quantitative CID-based identification of major sites of oxidation when sufficient MS/MS results were available. Most notably, we were able to confirm oxidation of W558, one of the two key tryptophan residues targeted for mutagenesis. Where available, this residue-level data has been plotted on the new Fig. S8.

8. To follow up with previous comment, it will be helpful if the authors can present a representative extracted ion

chromatogram for the wildtype and the hydroxyl radical modified species with corresponding MS/MS spectrum to justify their argument.

Reply: We have included all EICs of the unoxidized and all quantified oxidation products for all oxidized peptides, now included as Fig S9. Where available, we describe the residue-level annotation for each EIC.

9. Can the authors comment on the sequence coverage of the FPOP-MS analysis? If the sequence coverage is not (near) 100%, how did the authors rule out the possibilities that other regions that are not covered may also involve in binding?

Reply: Even after thorough optimization of digestion, we were unable to generate near 100% sequence coverage with peptides of usable size. This is not unusual in our experience for GAG-binding proteins, where the stretches of basic residues often render trypsin unsuitable. We have included both sequence coverage and label coverage in the new Fig S8. We have also noted in the figure legend that no topographical information can be inferred for regions that either showed no oxidation or that were not covered in our sequence coverage.

10. Lines 254-256, authors compared the melting curves for both wildtype and the mutant protein and emphasized that a clear transition from folded to unfolded was observed for the wildtype protein. Correct me if I'm not understanding it properly, but a clear transition is observed for the mutant as well (Figure 3c). If so, why is the "clear transition for wildtype" worth emphasizing?

Reply: Thank you for raising this point, it is of course the point that the mutant still shows a clear unfolding transition – this is corrected in the manuscript.

11. Lines 264-266, authors mentioned that the docking model was obtained based on all structural and biochemical results. Authors also mentioned the experimental procedures of the docking in lines 388-392. Can authors expand the discussion on how the structural and biochemical results were utilized in docking? Specifically, how FPOP-MS data contribute to the docking process?

Reply: We have not used FPOP data information during the docking process, the docking was mainly guided by the positive charge on the molecule surface.

12. Lines 435-436, 1 mM adenine was mentioned twice.

Reply: We have corrected this error.

13. It seems that the pICS is a hydroxyl radical scavenger and the advantage of using adenine as a dosimeter is to make sure the effective radical concentration is comparable under both pICS-free and pICS-bound states. It will be helpful if the authors can show some evidence (e.g., dosimetry readings) demonstrating the comparable effective radical concentration is achieved, so that the comparison in Figure 3a is better justified.

Reply: We have included offline dosimetry readings and standard deviations in the method section of the manuscript to show that the radical yields were not significantly different.

Reviewers' Comments:

Reviewer #1:

Remarks to the Author:

Overall, this is a much improved version of the manuscript both in terms of the quality of the results and the overall written document. Taken together with the already published structures described by Ma et al provides a strong basis for going forward in understanding CS binding and the development of novel efforts to treat the disease. That said, there are still a number of areas where the study is still lacking in terms of scientific rigour, particularly in terms of the overall quality and completeness of the structure refinement and FPOP analysis. These weaknesses are however offset by the availability to the data provided by the authors following publication. This will certainly provide lively debate and help to propel this field forward.

Reviewer #2:

Remarks to the Author:

The authors have addressed all concerns and the manuscript is now suitable for publication.

Reviewer #3:

Remarks to the Author:

The authors have adequately addressed my concerns and the manuscript is improved. I recommend to publish it.

Reviewer #4:

Remarks to the Author:

The authors responsibly addressed most of my concerns and comments in the revision. I would recommend the manuscript to be published in Nat. Commun. after addressing these two minor points below.

1. Line 101 and 104, please change "CSA" into "CS" for consistency.
2. Thanks for including all the extracted ion chromatograms in the revision, as it is very helpful for the readers. Since the authors highlighted residue W588 in the revision, it will be helpful for the authors to present in the Supporting Information the MS/MS spectra for the wildtype peptide and the peptide with oxidized W588 to further support their claim.

Reviewer #4 (Remarks to the Author):

The authors responsibly addressed most of my concerns and comments in the revision. I would recommend the manuscript to be published in Nat. Commun. after addressing these two minor points below.

1. Line 101 and 104, please change "CSA" into "CS" for consistency.

Reply: Yes thank you this has been changed

2. Thanks for including all the extracted ion chromatograms in the revision, as it is very helpful for the readers. Since the authors highlighted residue W588 in the revision, it will be helpful for the authors to present in the Supporting Information the MS/MS spectra for the wildtype peptide and the peptide with oxidized W588 to further support their claim.

Reply: Yes we have added a new figure S10 with this supporting data

To the Editor:

Further we have modified the manuscript according to the editors recommendations as requested in the word file provided by the editor. Please see this document that includes our point to point comments.